# High mTOR activity is a hallmark of reactive natural killer cells and amplifies early signaling through activating receptors

Antoine Marçais[1,2,3,4,5]*, Marie Marotel[1,2,3,4,5], Sophie Degouve[1,2,3,4,5], Alice Koenig[1,2,3,4,5], Sébastien Fauteux-Daniel[1,2,3,4,5], Annabelle Drouillard[1,2,3,4,5], Heinrich Schlums[6], Sébastien Viel[1,2,3,4,5,7], Laurie Besson[1,2,3,4,5], Omran Allatif[1,2,3,4,5], Mathieu Bléry[8], Eric Vivier[9,10], Yenan Bryceson[6,11], Olivier Thaunat[1,2,3,4,5], Thierry Walzer[1,2,3,4,5]*

[1]CIRI, Centre International de Recherche en Infectiologie - International Center for Infectiology Research, Lyon, France; [2]Inserm, U1111, Lyon, France; [3]Ecole Normale Supérieure de Lyon, Lyon, France; [4]Université Lyon 1, Lyon, France; [5]CNRS, UMR5308, Lyon, France; [6]Centre for Hematology and Regenerative Medicine, Department of Medicine, Karolinska Institutet, Karolinska University Hospital Huddinge, Stockholm, Sweden; [7]Laboratoire d'Immunologie, Hospices Civils de Lyon, Centre Hospitalier Lyon Sud, Lyon, France; [8]Innate-Pharma, Marseille, France; [9]Aix-Marseille Université, CNRS, INSERM, CIML, Marseille, France; [10]APHM, Hôpital de la Timone, Service d'Immunologie, Marseille, France; [11]Broegelmann Research Laboratory, The Gades Institute, University of Bergen, Bergen, Norway

*For correspondence:
antoine.marcais@inserm.fr (AM);
thierry.walzer@inserm.fr (TW)

**Abstract** NK cell education is the process through which chronic engagement of inhibitory NK cell receptors by self MHC-I molecules preserves cellular responsiveness. The molecular mechanisms responsible for NK cell education remain unclear. Here, we show that mouse NK cell education is associated with a higher basal activity of the mTOR/Akt pathway, commensurate to the number of educating receptors. This higher activity was dependent on the SHP-1 phosphatase and essential for the improved responsiveness of reactive NK cells. Upon stimulation, the mTOR/Akt pathway amplified signaling through activating NK cell receptors by enhancing calcium flux and LFA-1 integrin activation. Pharmacological inhibition of mTOR resulted in a proportional decrease in NK cell reactivity. Reciprocally, acute cytokine stimulation restored reactivity of hyporesponsive NK cells through mTOR activation. These results demonstrate that mTOR acts as a molecular rheostat of NK cell reactivity controlled by educating receptors and uncover how cytokine stimulation overcomes NK cell education.
DOI: https://doi.org/10.7554/eLife.26423.001

## Introduction

Natural killer (NK) cells are group 1 innate lymphoid cells characterized by their ability to kill target cells and to secrete cytokines such as IFN-γ (*Spits et al., 2013*). Thereby, they take part in the early response against infected and neoplastic cells. Target cell recognition and NK cell activation are controlled by the balance between positive and negative signals arising from the engagement of an array of NK activating receptors (NKar) and NK inhibitory receptors (NKir). While normal cells express an excess of NKir ligands, stressed cells, such as tumor and infected cells, may lose

**eLife digest** The cells of the immune system patrol the body to detect and destroy harmful microbes and diseased cells. Natural killer cells are immune cells with a natural capacity to kill infected or cancerous cells, as their name suggests. Importantly, they do so while sparing the surrounding healthy cells.

As natural killer cells mature they go through an "education" process to learn to distinguish between normal and abnormal cells. During education, the natural killer cells interact continuously with nearby healthy cells. However, it remains unknown how these interactions change the natural killer cells, or how these changes control their killing activity.

Marçais et al. now show that a protein called mTOR is essential to the education of natural killer cells. Comparing natural killer cells that had or had not completed the education process revealed that mTOR is more active in the educated cells. Moreover, inhibiting the activity of mTOR caused educated natural killer cells to lose their ability to identify diseased cells, while stimulating mTOR activity in uneducated natural killer cells mimicked the education process, allowing them to recognize and eliminate diseased host cells.

Certain nutrients are known to control the activity of mTOR, which suggests these nutrients could also affect how natural killer cells develop. In addition, manipulating the activity of mTOR could be used to control the response of natural killer cells to diseased host cells, and so could form part of treatments for cancer and infectious diseases. However, given that mTOR plays numerous roles within different body cells, any potential therapies that are developed would need to be able to manipulate mTOR specifically in natural killer cells.

DOI: https://doi.org/10.7554/eLife.26423.002

expression of NKir ligands ('missing-self') or acquire expression of NKar ligands ('modified-self'), thus tilting the balance towards activation. NKirs, which mostly recognize classical or non-classical MHC-I molecules, are stochastically expressed, resulting in a variegated expression pattern. Depending on the species, three types of NKir interact with MHC-I: Killer Immunoglobulin-like Receptors (KIR) in primates, Ly49 receptors of the lectin-like family in rodents and the heterodimer formed by CD94 and NKG2A in these species (*Vivier et al., 2008*).

In addition, considerable functional heterogeneity is observed in the NK cell population. Such cell intrinsic differences led to the proposition that NK cell reactivity and consequently their ability to discriminate self from non-self is the result of an education process (*Anfossi et al., 2006*; *Fernandez et al., 2005*; *Kim et al., 2005*). There was however considerable debate over the molecular process leading to education. Two theories were crafted to account for these observations: the first one proposing that a priming (or arming) signal was required to confer reactivity to otherwise hyporesponsive cells, the second positing that responsiveness is a default state that is lost upon unopposed chronic stimulation of NKar (disarming) (*Höglund and Brodin, 2010*). The data accumulated so far are in favor of the latter model, suggesting that intrinsic reactivity is lost upon chronic engagement of NKar unless this is opposed by concomitant engagement of NKir. Indeed, there is no evidence so far that priming signals are a prerequisite for acquisition of responsiveness. In contrast, disarming is the simplest explanation to account for the tolerance to self of NK cells raised in a mosaic or chimeric environment (*Johansson et al., 1997*; *Wu and Raulet, 1997*). Moreover, the loss of reactivity consequent to exposure of NK cells to activating ligands functionally demonstrates the possibility to disarm reactive NK cells (*Oppenheim et al., 2005*; *Tripathy et al., 2008*).

At the molecular level, physical interaction between NKirs and their ligands is required to maintain responsiveness as (1) only NK cells expressing NKir engaged by MHC-I at the surface of surrounding cells are reactive and as (2) NK cells expressing NKirs but developing in MHC-I deficient humans or animals are functionally impaired (*Fernandez et al., 2005*; *Kim et al., 2005*; *Zimmer et al., 1998*). In addition, the inhibitory signaling module acting downstream of NKirs is required to maintain reactivity. Indeed, mutation of the immunoreceptor tyrosine-based motifs (ITIM) of inhibitory Ly49 molecules or deficiency in the phosphatase SHP-1, recruited to NKirs upon ligation, decreases responsiveness (*Kim et al., 2005*; *Viant et al., 2014*). Inhibition of the activating signal by NKir thus serves two-distinct but related purposes: it counters inappropriate NK cell

activation and it prevents the desensitization induced by chronic stimulation thereby preserving NK cell reactivity. In inbred C57BL/6 mice, Ly49C (specific for H2-K$^b$), Ly49I (specific for H2-K$^b$) and the CD94/NKG2A receptor (specific for a D$^b$ peptide presented by Qa-1) have been shown to interact with substantial affinity with self-MHC class I molecules, while other receptors show no or marginal affinity (*Hanke et al., 1999*; *Michaëlsson et al., 2000*; *Vance et al., 1998*). Consequently, NK cell populations expressing these receptors are educated in C57BL/6 mice, that is, they are more reactive than their non-educated counterparts (*Fernandez et al., 2005*; *Joncker et al., 2009*; *Kim et al., 2005*). Education is a dynamic process tuned by the number of engaged NKirs and the strength of each interaction in a rheostat-like manner (*Brodin et al., 2009a*; *Johansson et al., 2005*; *Joncker et al., 2009*). It is also reversible in as little as one or two days as shown in different experimental set-ups (*Ebihara et al., 2013*; *Elliott et al., 2010*; *Joncker et al., 2010*). This suggests the existence of a potent cellular process integrating activating and inhibitory educating signals of variable strength (i.e. the strength of the NKar or NKir-ligand interaction and number of different interactions over time) and controlling the display of effector functions in response to NKar stimulation.

Previous studies have shown that reactive NK cells are characterized by stronger calcium flux and LFA-1 integrin activation upon NKar stimulation (*Guia et al., 2011*; *Thomas et al., 2013*). However, the nature of the molecular process conditioning NK cell reactivity and negatively affected by chronic engagement of NKar is unknown. To address this question, we systematically compared phosphorylation levels of key molecules involved in immunoreceptor tyrosine-based activating motif (ITAM) signaling in reactive vs. hyporesponsive NK cells at steady-state and following NKar stimulation. We discovered that NK cell reactivity is associated with a higher basal activity of the mammalian target of rapamycin (mTOR) pathway. Our genetic and pharmacological approaches collectively demonstrate a prominent role of mTOR signaling in controlling steady-state NK cell responsiveness.

## Results

### Reactive NK cells display higher activity of the Akt/mTOR pathway at steady-state and following acute NKar engagement

Seeking to identify molecular pathways involved in NK cell education, we systematically screened the basal levels of 20 phosphorylations on 16 proteins involved in ITAM signaling between reactive and hyporesponsive NK cells by flow cytometry (complete list in *Table 1*). This flow-cytometry based approach allowed us to combine the advantages of single-cell analysis and comparison of equivalent cell subset thanks to electronic gating. In C57BL/6 mice, the main educating NKirs are NKG2A and Ly49C, defining four subsets of which the double-negative display the lowest, the double-positive the highest and the single positives an intermediate responsiveness (*Joncker et al., 2009*). We also analyzed *B2m*$^{−/−}$ NK cells that are uniformly unreactive. Most of these phosphorylations are developmentally regulated (*Figure 1—figure supplement 1*), thus, to exclude any developmental bias, we compared similar developmental stages defined by CD11b and CD27 (*Figure 1—figure supplement 2*). Strikingly, all analyzed phosphorylations in the Akt/mTOR pathway correlated positively with the level of NK cell reactivity (*Figure 1A*). This was true when comparing C57BL/6 and *B2m*$^{−/−}$ NK cells as well as reactive and unreactive populations in C57BL/6 mice, regardless of the maturation stage. In C57BL/6 populations, absence of either NKG2A or Ly49C had a measurable negative effect, the absence of both leading to further decrease in the phosphorylation level. We also noted a significant correlation between education status and the level of pNFκB S529 and S468 as well as pLck Y505 and pItk Y180 (*Figure 1A*). However, as the most consistent differences lied in the Akt/mTOR pathway, we decided to focus our analysis on this pathway.

The phosphatase SHP-1 is required to maintain an optimal NK cell reactivity (*Viant et al., 2014*). To test its involvement in the maintenance of the basal activity of the Akt/mTOR pathway, we measured the phosphorylation levels of the ribosomal S6 protein and Akt in NK cells deficient in *Ptpn6*, the gene encoding SHP-1. As a control, we also measured the level of phosphorylation of STAT5 in these cells. The basal activity of the Akt/mTOR pathway was specifically decreased in NK cells from *Ncr1*$^{iCre/+}$ *Ptpn6*$^{lox/lox}$ mice compared to control NK cells while pSTAT5 levels were unchanged (*Figure 1B*). Thus, basal activation of the Akt/mTOR pathway is correlated with NK cell reactivity and controlled by SHP-1-dependent signaling downstream of NKirs.

**Table 1.** List of the antibodies used in this study and the phosphoepitopes they recognize.

| Phosphoepitope | Clone (Supplier) |
| --- | --- |
| pCD3ζ (Y142) | K25-407.69 (BD) |
| pLck (Y505) | 4/LCK-Y505 (BD) |
| pSyk (Y342) | I120-722 (BD) |
| pSLP76 (Y128) | J141-668.36.58 (BD) |
| pItk (Y180) | N35-86 (BD) |
| pPLCg2 (Y759) | K86-689.37 (BD) |
| pWIP (S478) | K32-824 (BD) |
| p-p38 (T180/Y182) | 36/p38 (pT180/pY182) (BD) |
| pERK1/2 (T203/Y205) | 20A (BD) |
| p-c-Cbl (Y698) | 47/c-Cbl (BD) |
| pJNK (T183/Y185) | N9-66 (BD) |
| pNFkB p65 (S468) | #3039 (CST) |
| pNFkB p65 (S529) | K10-895.12.50 (CST) |
| pNFkB p65 (S536) | 93H1 (CST) |
| pAkt (T308) | C31E5E (CST) |
| pAkt (S473) | M89-61 (BD) |
| pS6 (S235/236) | D57.2.2E (CST) |
| p4EBP1 (T36/45) | 236B4 (CST) |
| p-mTOR (S2448) | D9C2 (CST) |
| p-mTOR (S2481) | #2974 (CST) |

DOI: https://doi.org/10.7554/eLife.26423.007

We next compared mTOR-related signaling events arising from NKar stimulation in reactive versus hyporesponsive NK cells. To this end, we stimulated splenocytes from C57BL/6 (around 85% of NK cells are reactive in these mice) and $B2m^{-/-}$ mice by crosslinking NK1.1 and we measured phosphorylation events over time. Phosphorylation of Akt on T308 and S473 and phosphorylation of the ribosomal protein S6 were induced at higher levels in reactive NK cells compared to hyporesponsive NK cells (*Figure 1C,D*). By contrast, other signaling events not linked to the mTOR pathway were induced at similar levels (*Figure 1C,D* and *Figure 1—figure supplement 3*).

In summary, high activity of the Akt/mTOR pathway is a hallmark of reactive NK cells both at steady-state and following stimulation through NKars. Importantly, considering that education is not a discrete but rather a continuous process, absence of one or two of the educating NKir in C57BL/6 resulted in a commensurate loss in mTOR activity.

### Chronic NK cell stimulation results in decreased phosphorylation of the Akt/mTOR pathway which parallels the loss of reactivity

Education is rapidly reverted by unopposed chronic stimulation. Indeed, transfer of reactive NK cells into a host devoid of MHC-I leads to their rapid loss of reactivity and to their tolerance to MHC-I negative cells (*Joncker et al., 2010*). We thus sought to test whether chronic NKar stimulation decreased the activity of the Akt/mTOR pathway in parallel with the decrease of reactivity. To this purpose, we transferred reactive C57BL/6 NK cells into control C57BL/6 or $B2m^{-/-}$ mice and measured basal Akt/mTOR phosphorylation levels and their reactivity 3 days after transfer. To quantify the intensity of NKar signaling, we took advantage of a transcriptional reporter of the TCR signaling (*Moran et al., 2011*). This reporter consists of a GFP under the control of the promoter sequence of *Nur77*, an orphan nuclear receptor strongly induced in response to TCR stimulation. The signaling pathways triggered by TCR or NKar engagement mobilizing the same signaling adaptors, we reasoned that the $Nur77^{GFP}$ construct might also report NKar triggering. Indeed, in vitro stimulation with an NK1.1 agonist antibody or YAC-1 cells, a lymphoblastic cell line detected as foreign by C57BL/6 NK cells, resulted in an increase in the GFP fluorescence (*Figure 2—figure supplement 1*).

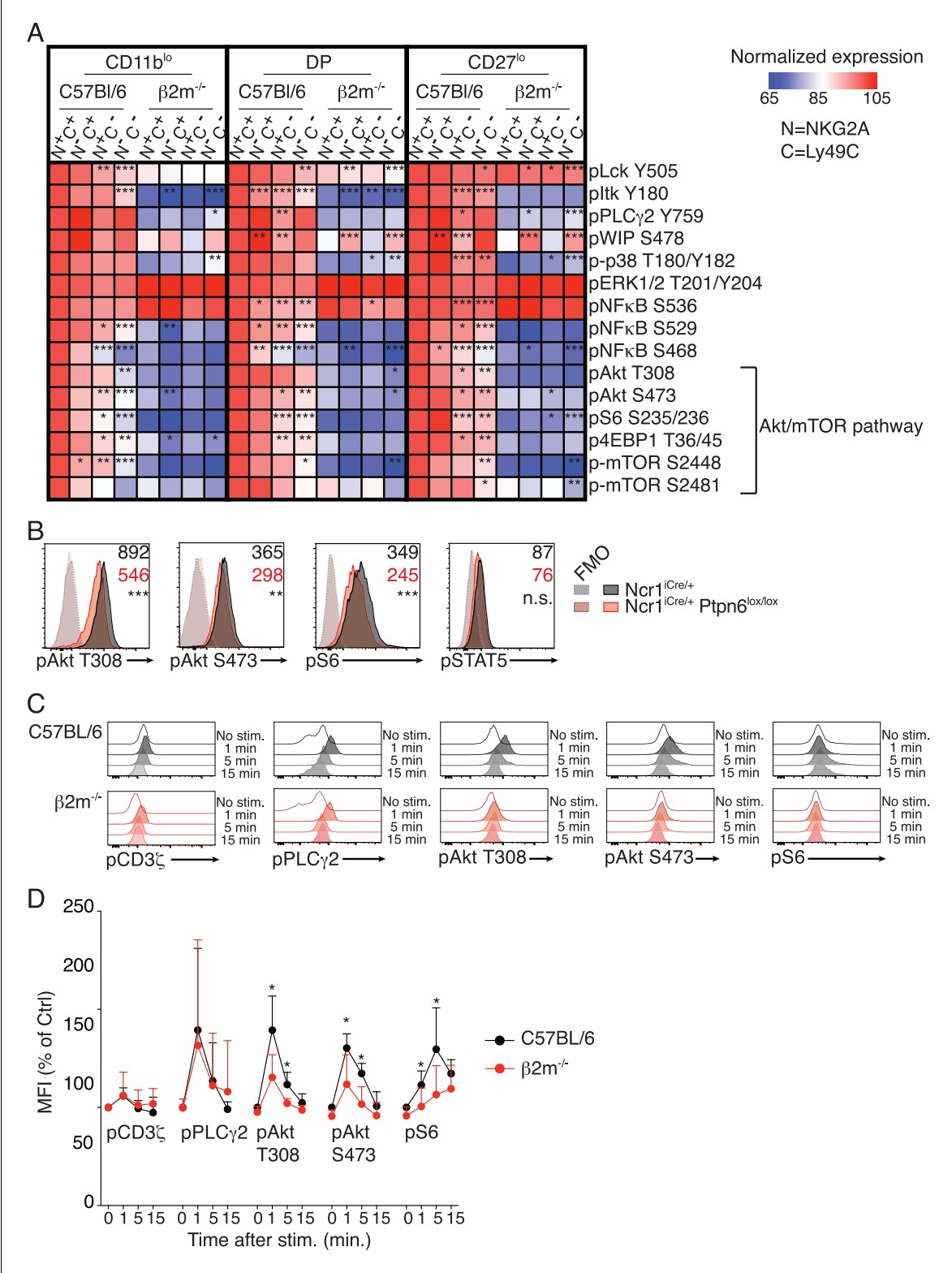

**Figure 1.** Basal activity of the mTOR pathway is proportional to the level of NK cell reactivity, and dependent on SHP1. (**A**) Heatmap representing the phosphorylation level of the phosphoepitopes indicated on the right in the different subsets of splenic resting NK cells indicated on top and gated as defined in *Figure 1—figure supplement 2*. Mean Fluorescence Intensity was recorded for each phosphoepitope in each subset. Normalized expression was calculated using the N$^+$C$^+$ subset of C57BL/6 mice as reference, as described in the Materials and Methods. The mean values are shown (n = 6 mice of each genotype in three independent experiments, adjusted p-values were calculated as described in the Materials and methods

*Figure 1 continued on next page*

*Figure 1 continued*
and compare the N+C+ subset to the indicated subset, *p<0.05, **p<0.01, ***p<0.001). (B) Histograms representing the phosphorylation level of the indicated proteins in splenic resting NK cells from *Ncr1*[iCre/+] or *Ncr1*[iCre/+] *Ptpn6*[lox/lox] mice (representative of 5 mice of each genotype in three independent experiments, t-test, **p<0.01; ***p<0.001, n.s. non significant). The MFI are indicated, in black for the *Ncr1*[iCre/+] NK cells and in red for the *Ncr1*[iCre/+]*Ptpn6*[lox/lox] NK cells. (C–D) Phosphorylation level of the indicated phospho-epitope in splenic NK cells from C57BL/6 or *B2m*[−/−] mice following NK1.1 stimulation for the indicated time. (C) Histogram overlays from one representative experiment. (D) MFI of the indicated phospho-epitope (mean +SD) of 5 mice of each genotype in five independent experiments (t-test, *p<0.05).
DOI: https://doi.org/10.7554/eLife.26423.003

The following figure supplements are available for figure 1:

**Figure supplement 1.** Bar graph showing the phosphorylation level of the indicated phosphoepitopes in the different subsets of splenic resting C57BL/6 NK cells defined by their expression of CD27 and CD11b.
DOI: https://doi.org/10.7554/eLife.26423.004

**Figure supplement 2.** Flow cytometry density plots presenting the analysis strategy to compare educated versus uneducated NK cells in C57BL/6 mice and the phenotypically equivalent subsets in *B2m*[−/−] mice.
DOI: https://doi.org/10.7554/eLife.26423.005

**Figure supplement 3.** Phosphorylation level of phospho-epitopes defined in *Table 1* was measured by flow-cytometry in splenic NK cells from C57BL/6 or *B2m*[−/−] mice following NK1.1 stimulation for the indicated time.
DOI: https://doi.org/10.7554/eLife.26423.006

Moreover, this increase was commensurate with reactivity so that higher GFP levels were reached in reactive NKG2A+Ly49C+ NK cells, thus validating the expression of GFP as a reporter of NKar stimulation. Transfer of *Nur77*[GFP] cells into *B2m*[−/−] mice resulted in a transient increase in the GFP level in the reactive subsets one day after transfer indicative of ongoing NKar signaling (*Figure 2A*). Interestingly, this was followed, 3 days after transfer, by a significant decrease in steady-state GFP level indicative of a loss of the cell capacity to signal following NKar stimulation. As previously reported, NK cells transferred into *B2m*[−/−] mice lost their reactivity while reactivity was maintained upon transfer into C57BL/6 host (*Figure 2B*, anti-NK1.1 stimulation and *Figure 2—figure supplement 2*, anti-NKp46 or YAC1 stimulation). Importantly, this was paralleled by a decrease in the phosphorylation of S6 and Akt S473 and a loss of the gradient observed between the different subsets expressing Ly49C and NKG2A (*Figure 2C*).

Collectively, these results demonstrate that the basal activity of the Akt/mTOR pathway is negatively affected by persistent and unopposed NKar stimulation. This suggests that engagement of Ly49C and NKG2A in C57BL/6 mice preserves Akt/mTOR basal activity resulting in higher basal phosphorylation in the NK cell population expressing these NKir.

## mTOR is essential for NK cell reactivity

To test if high mTOR activity was required for NK cell reactivity, we stimulated NK cells from *Ncr1*[iCre/+] *Mtor*[lox/lox] or control mice with plate-bound anti-NK1.1 antibody or YAC-1 cells and measured NK cell degranulation relative to the expression of the major educating receptors Ly49C and NKG2A. Control NK cells responded significantly better than mTOR-deficient NK cells, irrespective of the subset analyzed (*Figure 3A*). Moreover, within control NK cells, reactive Ly49C+NKG2A+ degranulated more than the other subsets, while mTOR deficiency resulted in equally hyporesponsive subsets.

These results suggested a major role of mTOR in NK cell reactivity. However, mTOR deficiency leads to a severe NK cell developmental block that may confound the interpretation of the results (*Marçais et al., 2014*). To address this issue we took advantage of Torin2, a highly selective ATP-competitive mTOR inhibitor targeting both mTORC1 and mTORC2 (*Liu et al., 2011*). We stimulated mature NK cells from C57BL/6 and *B2m*[−/−] mice with plate-bound anti-NK1.1 in the presence or absence of the inhibitor. Torin2 significantly decreased the capacity of C57BL/6 NK cells to produce IFN-γ and to degranulate upon stimulation, regardless of the subset analyzed (*Figure 3B*). Moreover, treatment of C57BL/6 NK cells with Torin2 abrogated the differences between highly reactive (Ly49C+NKG2A+) and hyporesponsive (Ly49C-NKG2A-) cells. Treatment of hyporesponsive *B2m*[−/−] NK cells led to a further decrease in their capacity to degranulate while their production of IFN-γ was unaffected. Similar results were obtained upon NKp46 stimulation (*Figure 3—figure supplement 1*). Torin2 treated C57BL/6 NK cells thus functionally behaved like *B2m*[−/−] hyporesponsive NK

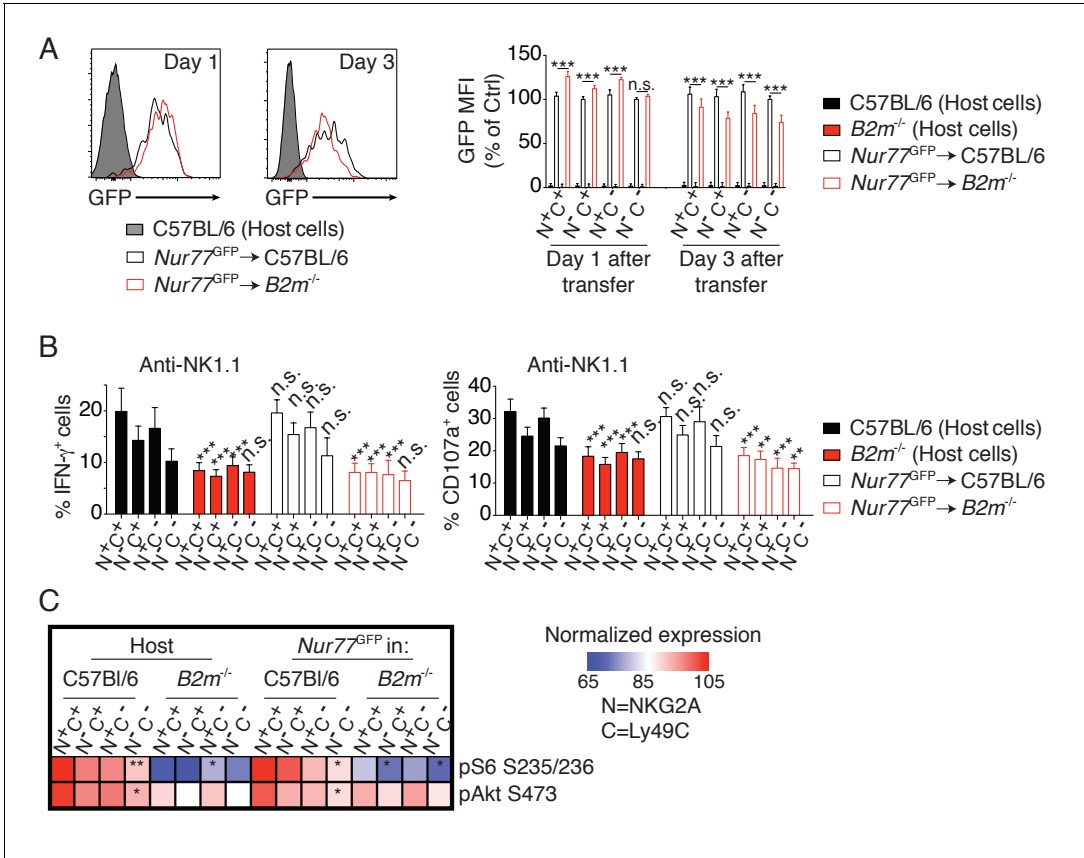

**Figure 2.** Reversion of education is accompanied by loss of the basal activity of the mTOR pathway. (**A**) *Left:* Representative histograms showing the GFP fluorescence levels of *Nur77*$^{GFP}$ NK cells transferred into C57BL/6 or *B2m*$^{-/-}$ mice and harvested 1 or 3 days after transfer. Non-transgenic host cells are shown. *Right:* Bar graph showing the GFP levels of the indicated splenic NK cell subsets normalized to the N$^-$C$^-$ population of *Nur77*$^{GFP}$ NK cells transferred into C57BL/6 control mice (mean +SD, n = 6 mice of each genotype per time point in two independent experiments, two-way ANOVA; ***p<0.001, n.s. non significant). (**B**) Percentage (mean + SD) of IFN-γ$^+$ or CD107a$^+$ cells among splenic host or transferred NK cells of the indicated subset following a 4 hr stimulation with coated anti-NK1.1. The experiment was done 3 days after transfer (n = 6 mice of each genotype in two independent experiments, two-way ANOVA comparing each subset to its counterpart in C57BL/6 mice, **p<0.01, ***p<0.001, n.s. non significant). (**C**) Heatmap representing the phosphorylation level of the phosphoepitopes indicated on the right in the different subsets of splenic resting NK cells indicated on top. Mean Fluorescence Intensity was recorded for each phosphoepitope in each subset. Normalized expression was calculated using the N$^+$C$^+$ subset of C57BL/6 host NK cells as reference. The mean values are shown (n = 6 mice of each genotype in two independent experiments, t-tests comparing the N$^+$C$^+$ subset to the indicated subset, *p<0.05, **p<0.01).

DOI: https://doi.org/10.7554/eLife.26423.008

The following figure supplements are available for figure 2:

**Figure supplement 1.** *Left:* Representative histograms showing the GFP fluorescence levels of *Nur77*$^{GFP}$ NK cells before or after a 4 hr stimulation with anti-NK1.1 or YAC-1 cells.

DOI: https://doi.org/10.7554/eLife.26423.009

**Figure supplement 2.** Percentage (mean +SD) of IFN-γ$^+$ or CD107a$^+$ cells among splenic host or transferred NK cells of the indicated subset following 4 hr stimulation with (**A**) coated anti-NKp46 or (**B**) YAC-1 cells.

DOI: https://doi.org/10.7554/eLife.26423.010

cells. Similarly, Torin2 inhibited C57BL/6 NK cells from triggering YAC-1 lysis at a similar level seen in hyporesponsive *B2m*$^{-/-}$ NK cells (*Figure 3C*). Torin2 treatment had no effect on the lytic capacity of *B2m*$^{-/-}$ NK cells.

Education conditions the phenomenon of missing-self recognition. A classical readout to highlight this property is to measure the rate of rejection of MHC-I negative target cells in vivo. To test whether basal activity of the Akt/mTOR pathway was involved in this process, we transferred a mix of C57BL/6 and NK-sensitive MHC-I negative (*B2m*$^{-/-}$) target cells into C57BL/6 mice, previously

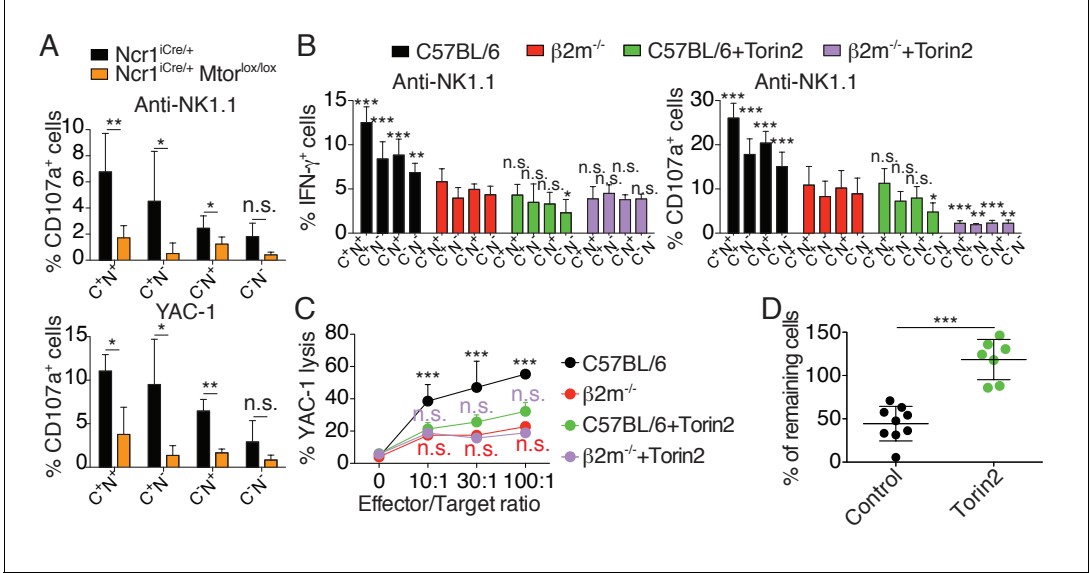

**Figure 3.** mTOR is essential for NK cell reactivity. (**A**) Percentage (mean +SD) of CD107a$^+$ cells among splenic CD11b$^{lo}$ NK cells of the indicated subset from *Ncr1*$^{iCre/+}$ or *Ncr1*$^{iCre/+}$ *Mtor*$^{lox/lox}$ mice following 4 hr stimulation with coated anti-NK1.1 or YAC-1 cells (n = 5 mice of each genotype in three independent experiments, t-tests comparing each subset in both genotype, *p<0.05, **p<0.01, n.s. non significant). (**B**) Percentage (mean +SD) of IFN-γ$^+$ or CD107a$^+$ cells among splenic NK cells of the indicated subset from C57BL/6 or *B2m*$^{-/-}$ mice following 4 hr stimulation with coated anti-NK1.1 in the presence or absence of 250 nM Torin2 (n = 9–10 mice of each genotype in five independent experiments, two-way ANOVA comparing each subset to its counterpart in *B2m*$^{-/-}$ mice, *p<0.05, **p<0.01, ***p<0.001, n.s. non significant). (**C**) Percentage (mean +SD) of dead YAC-1 cells after a 4 hr co-culture with purified NK cells of the indicated genotype at the indicated Effector/Target ratio in the presence or absence of 250 nM Torin2 (n = 9 C57BL/6 and 7 *B2m*$^{-/-}$ mice in four independent experiments, two-way ANOVA comparing each E/T ratio of C57BL/6 + Torin2 to C57BL/6, *B2m*$^{-/-}$ or *B2m*$^{-/-}$ +Torin2 as indicated by the color, ***p<0.001, n.s. non significant). (**D**) Percentage of remaining *B2m*$^{-/-}$ target cells following in vivo cytotoxicity experiment as described in the Materials and methods. Each dot represents a single mouse, bars indicate mean and SD (n = 9 control treated mice and 7 Torin2 treated mice in two independent experiments, t-test, ***p<0.001).

DOI: https://doi.org/10.7554/eLife.26423.011

The following figure supplement is available for figure 3:

**Figure supplement 1.** Percentage (mean +SD) of IFN-γ$^+$ or CD107a$^+$ cells among splenic NK cells of the indicated subset from C57BL/6 or *B2m*$^{-/-}$ mice following 4 hr stimulation with coated anti-NKp46 in the presence or absence of 250 nM Torin2 (n = 9–10 mice of each genotype in five independent experiments, 2-way ANOVA comparing each subset to its counterpart in *B2m*$^{-/-}$ mice, *p<0.05, **p<0.01, ***p<0.001, n.s. non significant).

DOI: https://doi.org/10.7554/eLife.26423.012

treated or not with Torin2. While injection into control mice led to the disappearance of 50% of the target cells, this rejection was abrogated in Torin2 treated animals, underlining the importance of mTOR activity in NK cell recognition of missing-self under steady-state conditions (*Figure 3D*).

Altogether, these results demonstrate that mTOR is required for NK cell reactivity.

## mTOR is a rheostat of NK cell reactivity through NKar

The 'rheostat' model of education proposes that the strength of the MHC-I input translates into a quantitative modification of NK cell responsiveness (*Brodin et al., 2009b*). Indeed, several studies reported that the higher the number of self–MHC-I receptors expressed by NK cells interacting with their ligands, the stronger their responsiveness (*Brodin et al., 2009a*; *Johansson et al., 2005*; *Joncker et al., 2009*). As shown in *Figure 1*, the level of mTOR activity was tightly correlated with the number of educating NKirs in NK cells, suggesting that mTOR could serve as the molecular rheostat translating the MHC-I input into quantitative tuning of the responsiveness. To directly test this point, we analyzed how the ex vivo modulation of mTOR activity by pharmacologic mTOR inhibitors changed NK cell responsiveness. We took advantage of four different inhibitors of graded mTOR inhibitory potential: the macrolide Rapamycin that primarily inhibits mTORC1 and three ATP-competitive inhibitors targeting both mTORC1 and mTORC2 to a varying extent: AZD2014, KU-0063794 (KU) and Torin2 (*García-Martínez et al., 2009*; *Guichard et al., 2015*; *Liu et al., 2011*;

*Sabatini et al., 1994*; *Yang et al., 2013*). The use of different concentrations of those compounds allowed us to modulate mTOR activity in NK cells over a dynamic range of 10-fold for mTORC1 or 2-fold for mTORC2 as measured by phosphorylation of S6 and Akt S473 respectively (*Figure 4A*). Of note, we confirmed that Rapamycin acted specifically on mTORC1 while AZD, KU and Torin2 inhibited both complexes. Importantly, at these concentrations no significant changes in STAT5 phosphorylation or specific toxicity over a 24 hr incubation period were noted (*Figure 4—figure supplement 1A and B*). We then correlated the S6 and Akt phosphorylation levels to the IFN-γ production and degranulation induced by NK1.1 crosslinking. S6 phosphorylation was positively correlated with the effector functions in all conditions tested (*Figure 4B*). Similar correlations were found between Akt phosphorylation and effector function upon AZD, KU or Torin2 treatment (*Figure 4C*). However, this correlation was lost upon Rapamycin treatment, suggesting that mTORC2 activity alone is not sufficient to sustain effector functions (*Figure 4B,C*). In addition, effector functions were not correlated to STAT5 phosphorylation levels (*Figure 4—figure supplement 1B,C*). Similar results were obtained upon stimulation of NK cells from *Ncr1*[iCre] and *Ncr1*[iCre] *Mtor*[lox/lox] mice and measure of the phosphorylation levels of the S6 and Akt proteins in parallel thus genetically confirming the results (*Figure 4—figure supplement 1D*).

Overall, these results demonstrate that mTOR acts as a molecular rheostat of NK cell responsiveness. Together with results in *Figures 1* and *2*, they demonstrate that NK cell education relies on the modulation of mTOR activity that in turn controls NK cell responsiveness through NKars.

## mTOR is essential for calcium response and integrin activation in NK cells following NKar engagement

Next, we asked whether mTOR activity could regulate signaling via NKar. Previous studies established that reactive NK cells display higher calcium flux (*Guia et al., 2011*) and higher integrin activation than hyporesponsive NK cells (*Thomas et al., 2013*). Hence we sought to test the impact of mTOR activity on these cardinal events in lymphocyte activation. We first measured the calcium flux in real time by flow cytometry following NK1.1 stimulation using fluorescent calcium probes and we quantified the intensity of the fluorescence peak. When we challenged *Ncr1*[iCre/+] (control) and *Ncr1*[iCre/+] *Mtor*[lox/lox] NK cells, NK1.1 cross-linking resulted in a detectable calcium flux in NK cells of both genotypes (*Figure 5A*). However, the peak was lowered (15–20%) in the absence of mTOR. We next applied the same protocol to control C57BL/6 NK cells in the presence or absence of Torin2 to acutely inhibit mTOR. As shown in *Figure 5B*, mTOR inhibition resulted in a decreased calcium flux characterized by a 20%-decrease in the peak intensity, thus phenocopying the impact of mTOR deficiency.

Next, we assessed the effect of mTOR deficiency on LFA-1 integrin activation following NKar triggering of inside-out signaling. For this purpose, we incubated NK cells from *Ncr1*[iCre/+] and *Ncr1*[iCre/+] *Mtor*[lox/lox] mice with beads coated with ICAM-1, the ligand of LFA-1, in the presence or absence of NK1.1 cross-linking. At different times, we measured by flow-cytometry the percentage of beads-associated NK cells as an indicator of LFA-1 activation in NK cells. As shown in *Figure 5C*, NK1.1 cross-linking failed to induce LFA-1 activation in mTOR-deficient NK cells contrary to control NK cells. In parallel, we also tested the effect of acute mTOR inhibition on LFA-1 activation in mature educated NK cells. As shown in *Figure 5D*, addition of Torin2 resulted in significant inhibition of LFA-1 activation induced by NK1.1 stimulation.

Thus, using genetic and pharmacological tools, we showed that the mTOR pathway lies upstream of two signaling events, calcium flux and LFA-1 integrin activation, which are elevated in reactive NK cells.

## Metabolic parameters of reactive and hyporesponsive NK cells

mTOR is a well-known regulator of the cell metabolism. We thus asked whether the higher activity of mTOR measured in reactive NK cells resulted in detectable changes in metabolic activity. We first measured cell size and granularity using the FSC and SSC flow-cytometry parameters. Reactive NK cells from C57BL/6 control mice presented a slight but significant increase of both morphological indicators when compared to hyporesponsive NK cells of *B2m*[−/−] mice (*Figure 6A*). Similarly, their mitochondrial content as well as glucose and fatty-acid uptake capacities estimated by measure of the uptake of the glucose fluorescent analog 2-NBDG or the fatty-acid fluorescent analog Bodipy

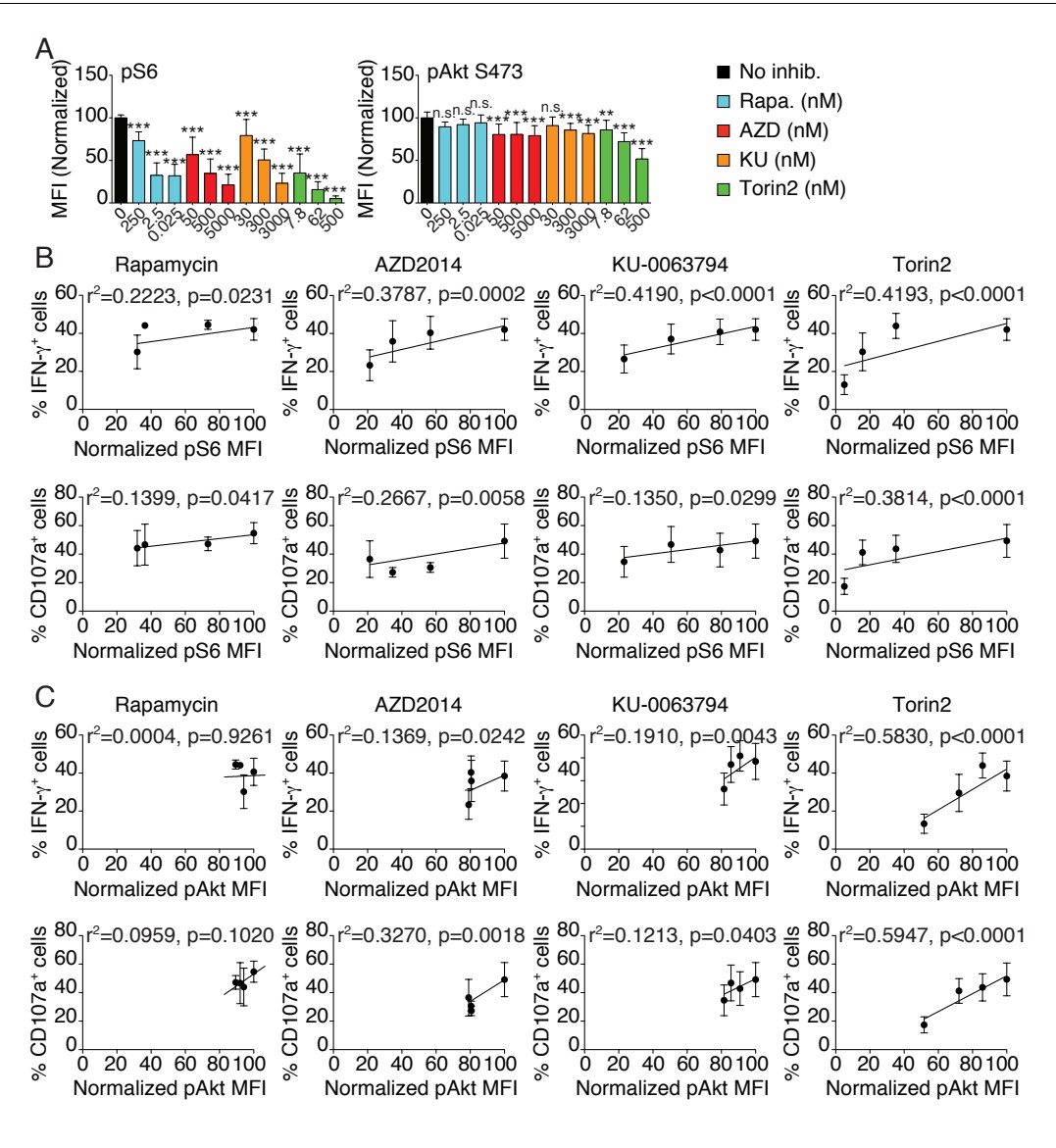

**Figure 4.** mTOR is a rheostat of NK cell reactivity through NKar. (A) Bar graph showing the phosphorylation level of S6 (left) and Akt S473 (right) in splenic NK cells following 1 hr treatment with 100 ng/ml IL-15 in the presence or absence of the indicated mTOR inhibitors at the indicated concentration (mean of the MFI normalized to the No inhibitor condition +SD, n = 9 mice in three independent experiments, one-way ANOVA comparing the No inhibitor condition with the indicated condition, **$p < 0.01$; ***$p < 0.001$, n.s. non significant). (B–C) Linear regression plots showing the correlation between (B) pS6 or (C) pAkt S473 as indicated and the percentage of IFN-$\gamma^+$ or CD107a$^+$ NK cells following 4 hr stimulation with coated anti-NK1.1 in the presence of 100 ng/ml IL-15 and mTOR inhibitors (mean ±SD, n = 9 mice in three independent experiments, the $r^2$ and p-value calculated by linear regression are indicated).

DOI: https://doi.org/10.7554/eLife.26423.013

The following figure supplement is available for figure 4:

**Figure supplement 1.** Bar graphs showing (A) the phosphorylation level of STAT5 in splenic NK cells following 1 hr treatment with 100 ng/ml IL-15 or (B) the percentage of live NK cells following a 24 hr culture in the presence or absence of the indicated mTOR inhibitors at the indicated concentration (A) mean of the MFI normalized to the No inhibitor condition or (B) percentage of live cells + SD, n = 9 mice in three independent experiments for pSTAT5 and 4 mice in two independent experiments for Viability, one-way ANOVA comparing the No inhibitor condition with the indicated condition, n.s. non significant).

DOI: https://doi.org/10.7554/eLife.26423.014

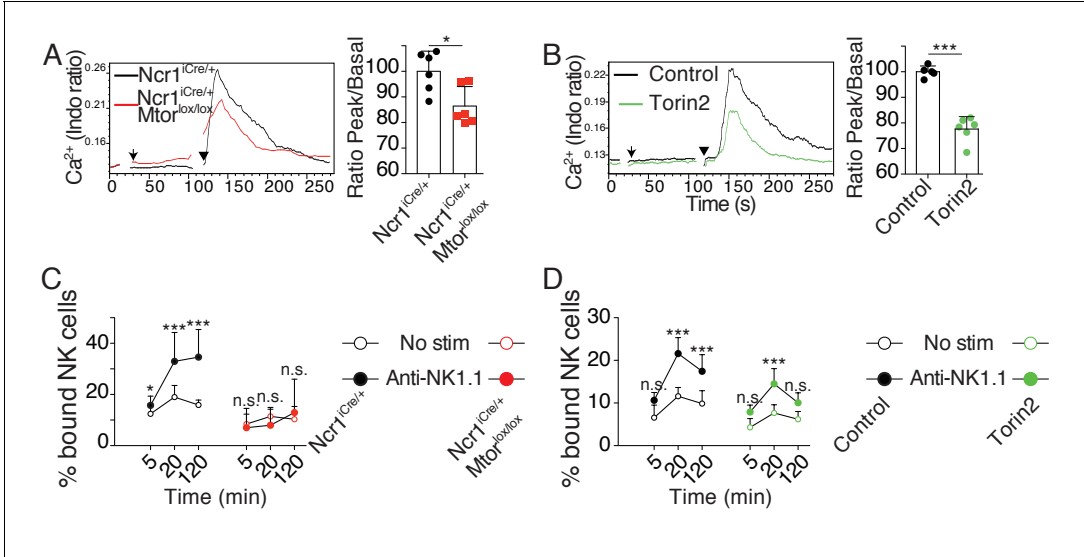

**Figure 5.** mTOR is essential for calcium response and integrin activation following NKar engagement. (**A**) *Left*: Representative histogram overlay showing the $Ca^{2+}$ flux intensity in splenic CD11b$^{lo}$ NK cells from *Ncr1*$^{iCre/+}$ or *Ncr1*$^{iCre/+}$ *Mtor*$^{lox/lox}$ mice. NK cells were activated following incubation with biotinylated anti-NK1.1 (Arrow) followed by cross-linking with streptavidin (Arrowhead). *Right*: Bar graph showing the Ratio Peak/basal normalized to the ratio of control NK cells (mean +SD, t-test *$p<0.05$). (**B**) *Left*: Representative histogram overlay showing the $Ca^{2+}$ flux intensity in splenic NK cells from C57BL/6 mice in the presence or absence of 500 nM Torin2. *Right*: Bar graph showing the Ratio Peak/basal normalized to the ratio of control NK cells (mean +SD, t-test ***$p<0.001$). (**C**) Percentage of splenic CD11b$^{low}$ NK cells from *Ncr1*$^{iCre/+}$ or *Ncr1*$^{iCre/+}$ *Mtor*$^{lox/lox}$ mice bound to beads coated with ICAM-1 after the indicated incubation time with or without NK1.1 stimulation (mean +SD, n = 6 mice of each genotype in four independent experiments, two-way ANOVA comparing the conditions with or without NK1.1 stimulation, n.s. non significant, **$p<0.01$, ***$p<0.001$). (**D**) Percentage of splenic C57BL/6 NK cells bound to beads coated with ICAM-1 after the indicated incubation time with or without NK1.1 stimulation, in the presence or absence of 250 nM Torin2 (mean +SD, n = 6 mice in four independent experiments, two-way ANOVA comparing the conditions with or without NK1.1 stimulation, n.s. non significant, ***$p<0.001$).

DOI: https://doi.org/10.7554/eLife.26423.015

FL-C16 were significantly higher (*Figure 6B*). In contrast, mitochondrial ROS production, lipid droplet content or lipid peroxidation were comparable in both cell types (data not shown). Differences were also detectable for FSC and SSC values as well as fatty-acid uptake when comparing reactive and hyporesponsive NK cell subsets present in the most mature CD27$^{low}$ population of C57BL/6 mice (*Figure 6C*).

In summary, the higher activity of the Akt/mTOR pathway observed in reactive cells increased their metabolic activity compared to hyporesponsive NK cells, which may also contribute to their enhanced responsiveness.

## Cytokine stimulation overcomes NK cell education by inducing high mTOR activity that restores NKar signaling

Several studies have demonstrated that hyporesponsive NK cells can be rendered reactive (*Ebihara et al., 2013*; *Elliott et al., 2010*; *Joncker et al., 2010*; *Sun and Lanier, 2008*). The underlying molecular mechanism has however remained elusive. We reasoned that if the mTOR pathway was really a key determinant of NK cell reactivity, acute activation of this pathway should immediately restore reactivity of hyporesponsive cells. To test this hypothesis, we stimulated NK cells from C57BL/6 or *B2m*$^{-/-}$ mice with plate-bound antibodies stimulating NK1.1 or NKp46 and we simultaneously added IL-2, a cytokine known to potently activate mTOR (*Marçais et al., 2014*). To test the requirement for the mTOR pathway in this process, cells were also treated or not with Torin2. IL-2 resulted in an increase of the cell capacity to produce IFN-γ and to degranulate as measured by CD107a exposure (*Figure 7A*). This acute treatment was sufficient for hyporesponsive cells to acquire a level of reactivity equal or even higher than that of reactive NK cells from C57BL/6, regardless of the stimulating antibody. mTOR activity was required for this effect since the increase in reactivity was suppressed by mTOR inhibition (*Figure 7A*). Similar results were obtained when using IL-

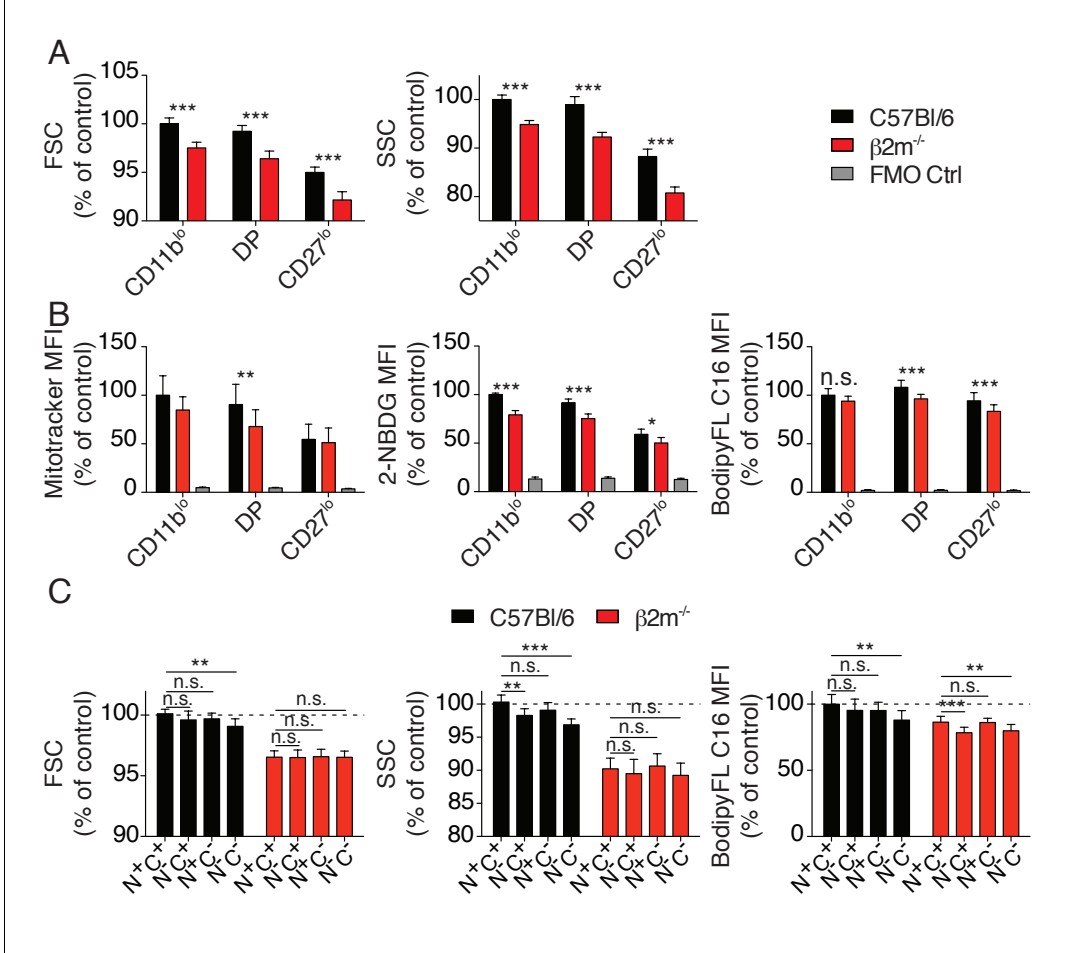

**Figure 6.** Metabolic parameters of reactive and hyporesponsive NK cells. (**A**) Bar graphs showing FSC and SSC values of splenic NK cell subsets from C57BL/6 or *B2m*<sup>−/−</sup> mice (mean +SD, n = 6 mice of each genotype in three independent experiments, t-test, ***p<0.001). MFI were normalized to the CD11b<sup>lo</sup> subset of C57BL/6 mice. (**B**) Bar graphs showing MFI of mitotracker staining, 2-NBDG or BodipyFL C16 incorporation of splenic NK cell subsets from C57BL/6 or *B2m*<sup>−/−</sup> mice (mean +SD, n = 6–10 mice of each genotype in three independent experiments, t-test, *p<0.05, **p<0.01, ***p<0.001, n.s. non significant). MFI were normalized to the CD11b<sup>lo</sup> subset of C57BL/6 mice. (**C**) Bar graphs showing FSC and SSC values or BodipyFL C16 incorporation of splenic NK cell subsets (gated on CD27<sup>low</sup>) from C57BL/6 or *B2m*<sup>−/−</sup> mice (mean +SD, n = 6–10 mice of each genotype in three independent experiments, t-test, n.s. non significant, *p<0.05, ***p<0.001). MFI were normalized to the N<sup>+</sup>C<sup>+</sup> subset of C57BL/6 mice.
DOI: https://doi.org/10.7554/eLife.26423.016

15 instead of IL-2 (*Figure 7—figure supplement 1*). Acute IL-15 stimulation also restored the cytotoxic activity of hyporesponsive NK cells against YAC-1 cells while further enhancing cytotoxicity of C57BL/6 cells (*Figure 7B*). Again, this effect was completely reversed upon concomitant Torin2 treatment. Taken together, these results show that induction of responsiveness in NK cells upon cytokine exposure is a rapid phenomenon acting via mTOR activation.

In order to decipher the mechanism required for NK cell re-education, we next tested whether acute IL-15 treatment restored early signaling in hyporesponsive cells. We first investigated the impact of IL-15 treatment on the calcium flux triggered by NK1.1 stimulation in control or hyporesponsive NK cells. As expected, NK1.1 stimulation of hyporesponsive NK cells resulted in a very poor calcium flux compared to reactive NK cells (*Figure 7C*). Strikingly, treatment with IL-15 increased the calcium flux ability of reactive and hyporesponsive NK cells in an mTOR-dependent way (*Figure 7C* and *Figure 7—figure supplement 2*). We then measured the impact of IL-15 treatment on LFA-1 activation following NK1.1 stimulation. The presence of IL-15 in the assay rendered hyporesponsive NK cells able to activate LFA-1 upon NK1.1 stimulation and bind ICAM-1 coated beads (*Figure 7D*).

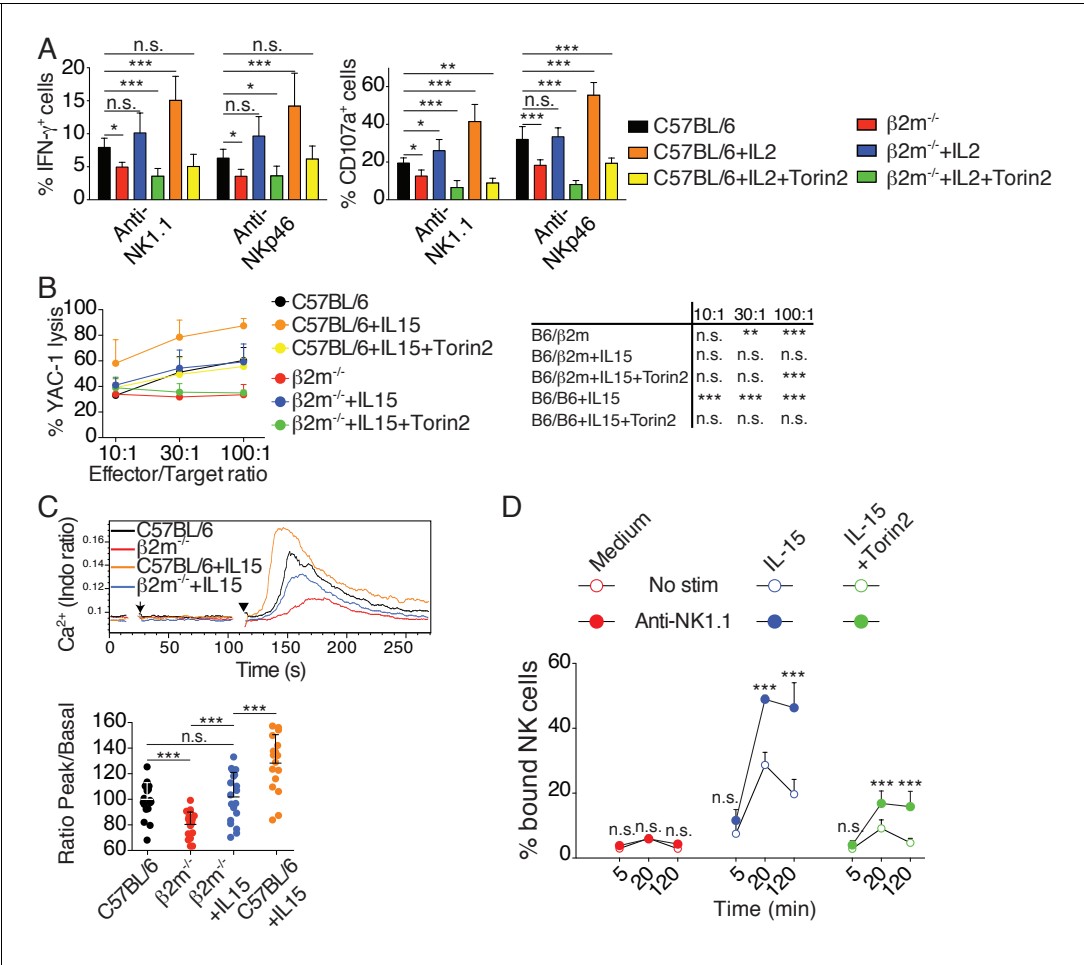

**Figure 7.** Cytokine stimulation overcomes NK cell education by inducing high mTOR activity that restores NKar signaling. (A) Percentage (mean +SD) of IFN-γ or CD107a positive cells among splenic NK cells from C57BL/6 or *B2m*$^{-/-}$ mice following 4 hr stimulation with coated anti-NK1.1 or anti-NKp46 in the presence or absence of 200UI/ml IL-2 and 250 nM Torin2 as indicated (n = 9–10 mice of each genotype in five independent experiments, one-way ANOVA comparing each condition to the C57BL/6 condition, *p<0.05, **p<0.01, ***p<0.001, n.s. non significant). (B) Percentage (mean +SD) of dead YAC-1 cells after a 4 hr co-culture with purified NK cells from C57BL/6 or *B2m*$^{-/-}$ mice at the indicated Effector/Target ratio in the presence or absence of 10 ng/ml IL-15 and 250 nM Torin2 as indicated (n = 7 mice of each genotype in three independent experiments, the table on the right presents the results of a two-way ANOVA comparing C57BL/6 with the other experimental conditions for the indicated Effector/Target ratio, **p<0.01, ***p<0.001 n.s. non significant). (C) *Top*: Representative histogram overlay showing the Ca$^{2+}$ flux intensity in splenic NK cells from C57BL/6 or *B2m*$^{-/-}$ mice with or without IL-15 (100 ng/ml). NK cells were activated following incubation with biotinylated anti-NK1.1 (Arrow) followed by cross-linking with streptavidin (Arrowhead). *Down*: Bar graph showing the Ratio Peak/basal normalized to the ratio of control NK cells (mean +SD of n = 17–20 replicates from 6 mice in six independent experiments, one-way ANOVA comparing the indicated conditions, **p<0.01, ***p<0.001). (D) Percentage of splenic NK cells from *B2m*$^{-/-}$ mice bound to beads coated with ICAM-1 after the indicated incubation time with or without NK1.1 stimulation, in the presence or absence of 100 ng/ml IL-15 and 250 nM Torin2 (n = 6 mice in four independent experiments, two-way ANOVA comparing stimulated to non-stimulated conditions, n.s. non significant, ***p<0.001).

DOI: https://doi.org/10.7554/eLife.26423.017

The following figure supplements are available for figure 7:

**Figure supplement 1.** Percentage (mean +SD) of IFN-γ or CD107a positive cells among splenic NK cells from C57BL/6 or *B2m*$^{-/-}$ mice following 4 hr stimulation with coated anti-NK1.1 or anti-NKp46 in the presence or absence of 10 ng/ml IL-15 and 250 nM Torin2 as indicated (n = 4 mice of each genotype in two independent experiments, one-way ANOVA comparing each condition to the C57BL/6 condition, *p<0.05, **p<0.01, ***p<0.001, n.s. non significant).

DOI: https://doi.org/10.7554/eLife.26423.018

**Figure supplement 2.** *Left*: Representative histogram overlay showing the Ca$^{2+}$ flux intensity in splenic NK cells from C57BL/6 or *B2m*$^{-/-}$ mice with or without IL-15 (100 ng/ml) and Torin2 (500 nM).

DOI: https://doi.org/10.7554/eLife.26423.019

This effect was strongly decreased upon Torin2 treatment, underlying the non-redundant role of mTOR in this process.

Altogether, these results show that acute stimulation of the mTOR pathway can restore the ability of hyporesponsive NK cells to induce calcium flux and activate LFA-1 upon NKar engagement, thereby re-establishing their reactivity.

## Discussion

Here, to gain mechanistic insight into the phenomenon of NK cell education, we explored signal transduction pathways downstream NKars in reactive and hyporesponsive NK cells. We found that the activity of the Akt/mTOR pathway was selectively higher in reactive NK cells. This was characterized by higher basal phosphorylation of direct and indirect targets of both mTOR complexes (mTORC1 and 2) in strict correlation with the reactivity level. In addition, this pattern was lost concomitantly with the loss of reactivity observed upon transfer of reactive cells in $B2m^{-/-}$ hosts. Our screen also revealed that two out of the three NFκB p65 phosphorylations investigated (S468 and S529) correlated with reactivity. This could be the result of the heightened Akt/mTOR pathway activity, as mTORC2 has been involved in NFκB activation during CD4 T cell stimulation (*Lee et al., 2010*). Alternatively, this could reveal the involvement of other pathways in the control of NK cell reactivity.

What is the extracellular signal and the signaling pathway responsible for the maintenance of mTOR basal activity in reactive NK cells specifically? An obvious candidate would be IL-15 as this cytokine is a privileged activator of this pathway (*Marçais et al., 2014*). However, pSTAT5 levels were identical between reactive and hyporesponsive NK cells (data not shown). Moreover, in vivo treatment with antibodies blocking IL-15 signaling did not alter NK cell education (data not shown). Finally, it is difficult to envisage how reactive cells would get preferential access to IL-15. Instead, in line with the disarming hypothesis, we would favor a model in which basal mTOR activity is set independently of education signals. This initial activity would then be decreased by disarming signals. How mTOR activity is decreased by chronic NKar stimulation is still an open question. We hypothesize that in the absence of surrounding MHC-I or in NK cells lacking functional NKirs, unopposed NKar signaling could lead to shut-down of the Akt/mTOR pathway due to depletion of necessary intermediates or establishment of negative feedbacks as it has been demonstrated in the case of induction of resistance to insulin (*Um et al., 2006*). Engagement of NKirs would prevent this desensitization and maintain an optimal activity of the pathway. In favor of this hypothesis, we show that SHP-1, the phosphatase triggered by NKir ligation and necessary to maintain NK cell reactivity (*Viant et al., 2014*), was required to maintain an optimal activity of the mTOR pathway. Furthermore, transfer of $Nur77^{GFP}$ cells into $B2m^{-/-}$ hosts was accompanied by an increase in the GFP level, evidence of active NKar signaling, and followed by the loss of mTOR basal activity concomitant with the loss of reactivity of NKG2A$^+$Ly49C$^+$ NK cells.

Previous studies have conclusively shown that NK cell education is not an on-off switch but rather a variation on a continuous axis (*Brodin et al., 2009a*; *Joncker et al., 2009*). We propose that the mTOR pathway is the long-sought molecular rheostat able to both respond to educating signals and control effector functions in return. Indeed, we showed that activity of the Akt/mTOR pathway is regulated commensurate with the level of NKir engagement by MHC-I molecules. Furthermore, we demonstrated that modulation of mTOR activity by exogenous cytokine or pharmacologic treatments was directly correlated with NK cell responsiveness. Furthermore, mTOR could also regulate NK cell responsiveness by integrating signals beyond NKir ligands. Considering the concept of the extended rheostat model as described initially by Höglund and colleagues (*Brodin et al., 2009b*), we envision extracellular inputs in an extended sense, including immunological as well as purely metabolic inputs. Interestingly, a number of environmental conditions, such as the presence of inflammatory (*Sun and Lanier, 2008*) or anti-inflammatory cytokines (*Sungur et al., 2013*), but also the presence of nutrients (*Keppel et al., 2015*), impact on NK cell responsiveness. All these stimuli positively or negatively affect mTOR activity (*Efeyan et al., 2015*; *Marçais et al., 2014*; *Sinclair et al., 2013*; *Viel et al., 2016*). mTOR activity could thus be the nexus targeted by these different stimuli which would explain their impact on NK cell responsiveness. Thus, considering mTOR as the rheostat of NK cell responsiveness would help to build a common conceptual framework in which these observations could be ordered.

Finally, we also present evidence on how mTOR activity affects NK cell effector functions. We demonstrated that mTOR activity controls two distinct events characterizing reactive NK cells and required for the triggering of effector functions: $Ca^{2+}$ flux and integrin activation. How could mTOR activity control such apparently unrelated signaling events? Depending on the relative involvement of mTORC1 or mTORC2, several possibilities can be considered. First, the fact that Rapamycin which specifically inhibits mTORC1 is sufficient to decrease responsiveness unmasks the non-redundant role of this complex. In line with the role of mTORC1 in the control of cellular metabolism, we described that higher basal mTOR activity in educated cells translated into higher basal metabolism as measured by morphological parameters as well as glucose and fatty-acid uptake and mitochondrial content. We and others have described the necessary role of the mTORC1-dependent metabolism in the development of NK cell effector functions (*Donnelly et al., 2014*; *Marçais et al., 2014*). In addition to improving the cellular fitness, metabolism could directly modulate signaling by controlling the availability of key intermediates as recently described for Th17/Treg differentiation (*Araujo et al., 2017*). Another possibility would be through the regulation of the actin cytoskeleton. Indeed, an emerging mode of lymphocyte signaling regulation is through cytoskeleton-dependent regulation of membrane receptors compartmentalization (*Mattila et al., 2016*), a process that has been proposed to explain the reactivity of educated NK cells (*Guia et al., 2011*). mTORC2 has been shown to regulate the cytoskeletal organization (*Huang et al., 2013*; *Sarbassov et al., 2004*) and could therefore prime reactive NK cells by cytoskeletal modifications. An interesting parallel can also be drawn with T cell anergy. Indeed, TCR stimulation in the absence of CD28 co-stimulation results in T cell hyporesponsiveness to further re-stimulation. Numerous studies have shown that the precise control of mTOR activity is at the heart of this phenomenon (*Chappert and Schwartz, 2010*; *Marcais et al., 2014*; *Zheng et al., 2007*; *2009*). Interestingly, this state is characterized by defective $Ca^{2+}$ flux (*Dubois et al., 1998*). Further resembling hyporesponsive NK cells, treatment of anergic T cells with IL-2 restores their responsiveness, an event that relies on mTOR activation (*Dubois et al., 1998*; *Zheng et al., 2007*). $Ca^{2+}$ flux is classically triggered by $IP_3$-induced release of endoplasmic reticulum stores which, upon detection by the STIM1/2 sensors, leads to opening of the ORAI channels present on the plasma membrane and extracellular $Ca^{2+}$ entry (*Hogan and Rao, 2015*). In addition, an underestimated $Ca^{2+}$ store is the endo-lysosomal compartment (*Morgan et al., 2011*), which constitutes a further link with mTOR since mTORC1 is activated on the lysosomal surface and positively regulated by lysosomal nutrients (*Efeyan et al., 2015*) as well as by calcium release from lysosomal stores (*Li et al., 2016*). Concerning regulation of integrin activation, a putative link would be through the inhibition of GSK3β. Indeed, this kinase is inhibited by Akt following mTORC2 activation (*Hagiwara et al., 2012*), and a recent study showed that its inhibition leads to better ability of NK cells to form conjugate via integrin activation (*Parameswaran et al., 2016*). In addition, PKCθ, a target of mTORC2 (*Lee et al., 2010*), activates WIP via S488 phosphorylation in lymphocytes (*Fried et al., 2014*). Since a macro-complex involving WIP, WASp, actin and myosin IIa has been defined in NK cells (*Krzewski et al., 2006*), WIP activation could explain better interaction with ICAM-1-coated beads in our assay and ultimately better docking to target cell.

In summary, these findings identify the activity of the mTOR pathway as the molecular rheostat responsible for the control of basal NK cell reactivity in response to NKir ligation. In addition, this provides a molecular basis for a number of previous experiments showing that NK cell education can be overcome by cytokine treatment. Finally, our data underline the extreme versatility of the regulation of NK cell responsiveness and further point to mTOR as a valid target for the manipulation of NK cells for therapeutic purposes.

## Materials and methods

### Mice and adoptive transfers

Wild-type C57BL/6 mice were purchased from Charles River Laboratories (L'Arbresle). $B2m^{-/-}$ (*Koller et al., 1990*), $Ncr1^{iCre/+}$ $Mtor^{lox/lox}$ (*Marçais et al., 2014*) and $Ncr1^{iCre/+}$ $Ptpn6^{lox/lox}$ mice (*Viant et al., 2014*) were previously described, littermate control mice were used as controls. $Nur77^{GFP}$ mice were previously described (*Moran et al., 2011*). Female mice 8 to 24 week-old were used. $Nur77^{GFP}$ splenocytes were injected i.v. in C57BL/6 or $B2m^{-/-}$ host. Each host received 25 × $10^6$ splenocytes labeled with CTV (1 μM, Molecular Probes) to allow subsequent identification. Host

mice were sacrificed one or 3 days after for analysis of the spleen. This study was carried out in accordance with the French recommendations in the Guide for the ethical evaluation of experiments using laboratory animals and the European guidelines 86/609/CEE. All experimental studies were approved by the bioethic local committee CECCAPP. Mice were bred in the Plateau de Biologie Expérimentale de la Souris, our animal facility.

## Flow cytometry

Single cell suspensions of spleens were obtained and stained. Intracellular stainings for phosphorylated proteins were done using Lyse/Fix and PermIII buffers (BD Bioscience). Measurement of glucose uptake was performed as described (*Marçais et al., 2014*). Mitochondrial content was measured using Mitotracker Green (Molecular Probes, 1 μM) incubated for 10 min at 37°C in PBS. Lipid uptake was measured using BodipyFL C16 (Molecular Probes, 1 μM) incubated for 30 min at 37°C in complete medium. Surface staining were then performed to identify the different populations. Flow cytometry was carried out on a FACS LSR II or on a FACS Fortessa (Becton-Dickinson). Data were analysed using FlowJo (Treestar). The following mAbs from eBioscience, BD Biosciences or Biolegend were used: anti-CD19 (ebio1D3), anti-CD3 (145–2 C11), anti-NK1.1 (PK136), anti NKp46 (29A1.4), anti-CD49b (DX5), anti-CD11b (M1/70), anti-CD27 (LG.7F9), anti-Ly49I (YLI90), anti-NKG2A/C/E (20d5), anti-IFN-γ (XMG1.2), anti-CD107a (1D4B). The mAb 4LO3311 recognizing Ly49C was purified on protein A column from supernatant of the 4LO3311 hybridoma generously provided by Pr. Suzanne Lemieux (Institut Armand Frappier, Québec). NKG2A positive cells were identified using the 20d5 clone which also recognizes NKG2C and NKG2E, however, since mouse resting NK cells only express NKG2A, we considered 20d5 reactive cells as NKG2A positive (*Vance et al., 1998*).

## Cell culture and stimulation

$1.5 \times 10^6$ splenocytes were cultured on antibody coated plates (anti-NKp46 (Goat polyclonal, R&D), anti-NK1.1 (PK136, BioXCell) at 10 μg/ml on Immulon 2HB or Nunclon plates) with Golgi-stop (BD Biosciences) in the presence of anti-CD107a for 4 hr. Cytokines and mTOR inhibitors were used at the following concentrations unless otherwise stated: rmIL-15 (Peprotech; 100 ng/ml), IL-2 (muIL-2 supernatant; 200 U/ml), Rapamycin (Calbiochem; 25 nM), KU-0063794 (Stemgent; 3 μM), AZD2014 (Selleckchem; 5 μM) and Torin2 (Tocris; 250 nM). Surface and intracellular stainings were then performed and IFN-γ production as well as CD107a exposure was measured by flow cytometry. In some experiments, cell viability was determined using 7AAD (Invitrogen, 250 nM).

For phospho-flow stainings following short-term NK1.1 stimulation, $3 \times 10^6$ splenocytes were stimulated using biotinylated NK1.1 (PK136, 5 μg/ml) followed 1 min 30 s later by streptavidin (Life Technologies, 10 μg/ml) and fixed by addition of 10 volumes of Lyse/Fix at the indicated time point.

## In vivo cytotoxicity assay

Recipient mice were treated by daily i.p. injection of Torin2 (10 mg/kg, vehicle: 40% H2O, 40% PEG400 (Sigma), 20 % N methyl two pyrrolidone (Sigma)) for 6 days prior to target transfer. Splenocytes from C57BL/6 or $B2m^{-/-}$ mice were labeled respectively with CellTraceViolet (1 μM) or CFSE (5 μM) (both from Life Technologies), and $10 \times 10^6$ cells ($5 \times 10^6$ of each genotype) were transferred by i.v. injection. 60 hr after transfer, splenocytes were isolated and analyzed by FACS. Percentage of remaining $B2m^{-/-}$ cells was calculated using the following formula: % remaining cells = 100 x (number $B2m^{-/-}$ cells/number C57BL/6 cells) at 60 h /(number $B2m^{-/-}$ cells/number C57BL/6 cells) in input mix.

## In vitro cytotoxicity assay

NK cells were first enriched by negative depletion prior to killing assay. Briefly, splenocytes suspension were incubated with biotinylated mAb against: CD3 (14–2 C11), TCRβ (H57-597), TCRγδ (GL3), CD19 (ebio1D3), TER-119 (ter119) (eBioscience), followed by incubation with anti-biotin microbeads (Miltenyi), and enrichment by magnetic separation on an AutoMACS. Enriched NK cells were co-cultured for 4 hr with YAC-1 cells labeled with CFSE (Life Technologies) at different Effector to target (E/T) ratios calculated based on the cell number and the percentage of NK cells after purification.

The percentage of dead cells within CFSE positive YAC-1 cells was measured by flow cytometry after staining with 7AAD.

## Calcium flux

Calcium flux was measured essentially as described (*Guia et al., 2011*). Briefly, RBC-lysed splenocytes suspension in RPMI/0.2% BSA/25 mM HEPES were stained at RT with the following mAb: anti-CD3/CD19 PEeFluor610, anti-CD49b APC, anti-CD11b APCCy7, anti-CD27 PE. They were then stained at $1 \times 10^7$ cells/ml with Indo-1 (1 μM, Life Technologies) for 30 min at 37°C and washed two times at 4°C. They were resuspended in the above medium and placed at 37°C for 30 min prior acquisition in the presence or absence of rmIL-15 (100 ng/ml) or Torin2 (250 nM). Samples were acquired on a LSRII (BD) as follow: 15 s baseline acquisition, addition of anti-NK1.1 biotin (PK136, 5 μg/ml), acquisition for 1 min 30 s, addition of Streptavidin (Life Technologies, 10 μg/ml) and, acquisition for another 3–5 min.

## ICAM1 coated beads assay

One mg Protein G-coated 4–4.9 μm beads (Spherotec) was incubated for 30 min with 3.5 μg ICAM1-hIgG1Fc (R&D) on a rotating wheel at RT in PBS. Beads were then pelleted by centrifugation and washed two times with complete medium, counted on a FACS Accuri (BD) and resuspended at $1 \times 10^7$ beads/ml. In parallel, NK cells were purified (80–90% purity) using biotinylated antibodies directed against CD3, CD19, CD5, CD24, F4/80 and Ly6G and anti-biotin beads. They were then incubated with anti-NKp46-PE (29A1.4, BD) and purified anti-NK1.1 (PK136, BioXCell). 100,000 purified NK cells in 10 μl were placed in a U-bottom well and 100,000 ICAM-1 coated beads were added. To cross-link NK1.1 and measure the effect of inside-out signaling, a Goat F(ab)′$_2$ anti-mouse IgG (10 μg/ml, Life Technologies) was added to the wells. Interaction was fixed at the indicated time-point by addition of 100 μl Cytofix/Cytoperm (BD). The percentage of interaction (i.e. percentage of NKp46 positive cells attached to beads) was measured by flow cytometry.

## Statistical analysis

Statistical analyses were performed using Prism 5 (Graph-Pad Software). Two tailed unpaired t-test, and ANOVA tests with Bonferroni correction were used as indicated in the figure legends. Significance is indicated as follows: *$p<0.05$; **$p<0.01$; ***$p<0.001$. The heatmap presented in *Figure 1A* was established as follow: we first selected the phosphoepitopes for which the MFI (Mean Fluorescence Intensity) was significantly above the one of the FMO control (Student T-test). The MFI of the 15 selected phosphoepitopes for the 4 NC sub-populations defined in *Figure 1—figure supplement 2* was then normalized to the MFI value of the NKG2A$^+$Ly49C$^+$ populations in the C57BL/6 mice and the values obtained were averaged to calculate the means for each populations. These values were used to establish the Heatmap using the Multiple Experiment Viewer application. We used the R statistical language to manage our database and carry out the statistical analysis (R version 3.3.2). We splited the database into six datasets (2 Mouse strains * Differentiation subsets), each containing the 15 phospho-epitopes. We performed an ANOVA for each phospho-epitope to test for the phosphorylation difference between the 4 NC sub-populations. The parameters of the ANOVA Type I SS were adapted to control for the experiment effect. The Bartlett Homogeneity of Variances Test was applied first, when it failed to reject its H0, then the phospho-epitope was retained for the ANOVA test. The normality of the residuals of the ANOVA model was checked graphically and numerically with the Shapiro-Wilk Normality Test. When this test failed to reject its H0 then the adjusted P values for multiple comparisons were extracted with the Tukey's 'Honest Significant Difference' method.

## Acknowledgements

The authors acknowledge the contribution of SFR Biosciences (UMS3444/CNRS, ENSL, UCBL, US8/INSERM) facilities, in particular the Plateau de Biologie Expérimentale de la Souris, and the flow cytometry facility. We would like to thank Pr. Suzanne Lemieux (INRS, Institut Armand-Frappier, Laval, Québec, Canada) for providing us with the 4LO3311 hybridoma and Pr. David Raulet for advice on its purification. We would like to thank Pr. Kristin Hogquist for the *Nur77*$^{GFP}$ mice. The TW lab is supported by the Agence Nationale de la Recherche (ANR JC *SPHINKS* to TW and ANR JC *BaNK* to AM), the ARC foundation (équipe labellisée), the European Research council (ERC-Stg

281025), and receives institutional grants from the Institut National de la Santé et de la Recherche Médicale (INSERM), Centre National de la Recherche Scientifique (CNRS), Université Claude Bernard Lyon1 and ENS de Lyon. MM is the recipient of a fellowship from La Ligue Nationale contre le Cancer.

## Additional information

### Competing interests

Mathieu Bléry: MB is employee of Innate-Pharma. Eric Vivier: EV is shareholder of Innate-Pharma. The other authors declare that no competing interests exist.

### Funding

| Funder | Grant reference number | Author |
|---|---|---|
| Agence Nationale de la Recherche | ANR-16-CE15-0005-01 Bank | Antoine Marçais |
| H2020 European Research Council | 281025 Dironaki | Thierry Walzer |

The funders had no role in study design, data collection and interpretation, or the decision to submit the work for publication.

### Author contributions

Antoine Marçais, Conceptualization, Formal analysis, Supervision, Funding acquisition, Validation, Investigation, Methodology, Writing—original draft, Writing—review and editing; Marie Marotel, Alice Koenig, Sébastien Viel, Formal analysis, Investigation, Writing—review and editing; Sophie Degouve, Formal analysis, Investigation, Methodology, Writing—review and editing; Sébastien Fauteux-Daniel, Annabelle Drouillard, Laurie Besson, Formal analysis, Investigation; Heinrich Schlums, Investigation, Methodology, Writing—review and editing; Omran Allatif, Formal analysis, Validation, Methodology; Mathieu Bléry, Eric Vivier, Resources, Writing—review and editing; Yenan Bryceson, Supervision, Validation, Writing—review and editing; Olivier Thaunat, Supervision, Validation, Methodology, Writing—review and editing; Thierry Walzer, Conceptualization, Supervision, Funding acquisition, Project administration, Writing—review and editing

### Author ORCIDs

Antoine Marçais (iD) https://orcid.org/0000-0002-3591-6268
Thierry Walzer (iD) http://orcid.org/0000-0002-0857-8179

### Ethics

Animal experimentation: This study was carried out in accordance with the French recommendations in the Guide for the ethical evaluation of experiments using laboratory animals and the European guidelines 86/609/CEE. All experimental studies were approved by the bioethic local committee CECCAPP (Permit number: CECCAPP_ENS_2014_018).

### Decision letter and Author response

Decision letter https://doi.org/10.7554/eLife.26423.021
Author response https://doi.org/10.7554/eLife.26423.022

## Additional files

### Supplementary files

• Transparent reporting form
DOI: https://doi.org/10.7554/eLife.26423.020

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
