## [Decision Letter]

[Editors’ note: this article was originally rejected after discussions between the reviewers, but the authors were invited to resubmit after an appeal against the decision.]

Thank you for submitting your work entitled "mTOR is a molecular rheostat integrating education signals to regulate Natural Killer cell responsiveness" for consideration by *eLife*. Your article has been reviewed by three peer reviewers, and the evaluation has been overseen by a Reviewing Editor and a Senior Editor. The reviewers have opted to remain anonymous.

Our decision has been reached after consultation between the reviewers. Based on these discussions and the individual reviews below, we regret to inform you that your work will not be considered further for publication in *eLife*.

Although your paper was viewed with great interest, I am afraid that the consensus of the reviewers and the Editors is that the main conclusions of the paper on NK cell education were not supported by the data presented. The reviewers also discussed the possibility that the manuscript could be sufficiently edited to support more limited conclusions. However, in that case, the consensus view was that the paper would then not be enough of an advance in the field to justify publication in *eLife*. The complete detailed reviewers of your manuscript are attached below, in the hopes that they will help you in revising your paper for publication elsewhere.

*Reviewer #1:*

The main conclusion of this paper, namely that mTOR is essential for NK cell education does not hold. In a nice set of experiments the authors show that mTOR contributes to activation signals in NK cells in response to various stimuli, including target cells and signaling by receptors NK1.1 and NKp46. To make such a conclusion it would be necessary to dissociate the role of mTOR in NK cell activation and education. For example, had mTOR acute inhibition by Torin not affected NK cell responses to in vitro stimuli, while deletion of mTOR in NK cells in mice still resulted in lack of education, one could then conclude that mTOR may have a role in education.

Cytokine stimulation, which activates the mTOR pathway, is taken as evidence for a role of mTOR in overcoming the weak responses of uneducated NK cells. This conclusion cannot be made on the basis of the evidence. It is known that IL-2 or IL-15 overcome NK unresponsiveness and stimulate the mTOR pathway. But these cytokines could overcome unresponsiveness independently of their stimulatory effect on the mTOR pathway. The fact that mTOR contributes to NK activation in the context of IL-2 or IL-15 stimulation does not imply a role in educating NK cells.

This is a nice study on the contribution of the mTOR pathway to NK cell activation both in vitro and for MHC-I negative tumor cell rejection in vivo. Using titrations of inhibitors the authors show that the degree of mTOR activity correlates well with the strength of NK responses, again showing that mTOR is required for NK activation. mTOR contributes to early steps of NK activation such as calcium responses and activation of integrin.

1) The main conclusion is not supported by the data, as described above.

2) There is no evidence so far that NK cell education by inhibitory receptors is due to a unique educating signal, a hypothesis known as the arming hypothesis. However, there is evidence that continuous stimulation of NK cells leads to hyporesponsiveness. Accordingly, the basis for education is that inhibitory signals prevent induction of hyporesponsiveness. The experiments in this paper should be interpreted in that context. The mTOR pathway could be a particularly sensitive target of activation-induced hyporesponsiveness. In the Introduction section the authors actually say that 'the current viewpoint is that engagement of the NKirs prevents the desensitization of the NK cell due to chronic NKar stimulation'. However, this viewpoint has not been considered in this paper, which is interpreted mainly according to the existence of educating signals.

3) The Discussion is interesting and well written. But it is based on the unsupported premise that mTOR is essential for NK cell education.

*Reviewer #2:*

In this manuscript, Yu et al. showed that increased basal activity of the mTOR/Akt pathway is associated with mouse NK cell education defined by expression of the inhibitory receptors. They also found that mTORC1 signaling is regulated by the SHP-1 phosphatase and is associated with the improved responsiveness of educated NK cells. They further showed that upon stimulation through activating NK cell receptors, the mTOR/Akt pathway enhances calcium flux and LFA-1 integrin activation. In accordance with published work, pharmacological inhibition of mTOR results in decreased NK cell reactivity, while acute cytokine stimulation restores responsiveness of uneducated NK cells through mTOR activation.

This manuscript studies the previous unaddressed role of mTOR/Akt pathway in NK cell education, and is well written. While most data supports the author's overall conclusion, additional experiments and discussion (as detailed below) are needed to clarify and support all the claims made.

Figure 1: What is the phosphorylated protein status between CD11b^lo^, DP, and CD27lo NK cells? What is the author's thought on the increased phosphorylation of ERK1/2 T201/Y204 and pNFkB S5356 in NK cells from B2m-deficient mice?

Figure 2: NK cells in Ncr1iCre Mtorfl/fl mice have severe developmental defects. Therefore, can the authors subset NK cells in this figure based on how they subset in Figure 1 (by inhibitor receptors and CD11b and CD27 expression level)?

Figure 3: The inhibitor used in this figure have different mechanisms of action; some of them inhibit only mTORC1 while some inhibitor both mTORC1 and mTORC2. Can they titrate the amount of the same inhibitor to get the graded pS6 level and compare that with IFN-γ production as well as degranulation?

Figure 6: This figure lacks the control of NK cells from WT B6 mouse treated with IL2 or IL15 with or without Torin2.

*Reviewer #3:*

This group previously published an interesting paper in Nature Immunology reporting that mTOR is required for IL-15 signaling in the development and activation of NK cells. The current manuscript shows that mTOR plays a role in NK cell education; this story is potentially interesting but moreexperiments are necessary to support conclusions.

1) In Figure 2, the authors should perform the stimulation of NK cells with anti-NKP46 to measure% IFNγ+ cells from Ncr1-cre+ and Ncr1-Cre-Mtor lox/lox.

2) In Figure 2, it is unclear how mTOR acts in *B2m^-/-^*NK cells. The authors should use as an additional condition: *B2m^-/-^*+Torin2. Since the authors have mtor conditional ko animals, it would be important to perform a parallel expt using *B2m^-/-^*X Ncr1-cre+Mtor lox/lox vs. Ncr1cre+Mtorl ox/lox vs. Ncr1cre+ NK cells.

3) The authors mentioned that rapamycin primarily inhibits mTORC1 which is true, but only using with a low concentration (e.g. 10 or 20 nM). Concentrations up to 100 nM inhibit both mTORC1 and mTORC2. In Figure 3, the authors used *extremely* high concentrations of rapalogs, which could lead to some artificial effects. The authors used "3 uM" of KU0063794 for treating NK cells and observed a significant defect of pS6 and pAKT S473. However, other groups have used "100 nM" of KU0063794 to investigate mTOR roles in dendritic cell activation (PMID: 22826320). Thus, authors need to show the cell viability data from those rapalog treatments. In addition, it is necessary for authors to demonstrate that Ncr1cre+Mtor lox/lox NK cells phenocopy the same results.

4) In Figure 5, mTOR regulates glucose metabolism in various cells, but authors only compared glucose uptake by using 2-NBDG from educated and non-educated NK cells; have the authors tried to look at glucose transporter 1 (Glut1) level? Also, it would be nice if the authors could explore in depth other metabolic parameters, for instance mitochondrial function lipid uptake and oxidation?

5) In Figure 6, it would be great to have data that NK cells from *B2m^-/-^*+Torin2 in the comparison. If possible it would be nice to see NK cells from *B2m^-/-^*X Ncr1cre+ Mtor lox/lox vs. *B2m^-/-^*X Ncr1cre+ vs. Ncr1cre+ animals.

[Editors’ note: what now follows is the decision letter after the authors submitted for further consideration.]

Thank you for choosing to send your work entitled "mTOR is a molecular rheostat integrating education signals to regulate Natural Killer cell responsiveness" for consideration at *eLife*. Your letter of appeal has been considered by a Senior Editor, a Reviewing editor, and all three reviewers, and we are prepared to consider a revised submission with no guarantees of acceptance.

Reviewer #1 had the following specific comments that should be addressed in the re-submission.

I appreciate the measured response from the authors and agree it would be fair to let them revise the paper in the way they have proposed. Let me comment on their response.

Their replies to specific points are preceded by five points taken out of my summary paragraph. Their reply to the first suggests that I have not delivered my main message clearly enough. The second main message they propose for the revised manuscript is that "education signals prime basal mTOR activity." In the NK field the concepts of priming and education are not well understood and their relationship is unclear. Priming as an early event not known to be reversible. In contrast, education is rapidly reverted by chronic stimulation. There is no evidence that education signals are a prerequisite for acquisition of responsiveness. Therefore, basal mTOR activity may be independent of education signals. However, mTOR activity could be lost or diminished by disarming. The authors should interpret their data within the context of two models: education signals that are required versus an intrinsic ability to respond via mTOR that can be switched off.

In the second point, I accept their argument on the role of mTOR in cytokine induced education.

On the fifth point, I agree that the role of mTOR in early NK activation signals is a new contribution.

Specific points: 1. As above. 2. This point relates to my first comment above. The authors have proposed an interesting experiment. >>>

Reviewer #2 had the following specific comment that should be addressed in the re-submission.

One thing that they still can’t quite explain is why there is increased phosphorylation of ERK1/2 T201/Y204 and pNFkB S5356 in NK cells from b2m-deficient cells. They said that it might be due to unopposed chronic stimulation through NKar…But then if this is the case, NKar also activated mTORC1… What is the difference in the regulation of these phosphorylation in NK cells or why chronic stimulation through NKar negatively impact mTOR signaling?

---

## [Author Response]

[Editors’ note: the author responses to the first round of peer review follow.]

Reviewer #1:The main conclusion of this paper, namely that mTOR is essential for NK cell education does not hold. In a nice set of experiments the authors show that mTOR contributes to activation signals in NK cells in response to various stimuli, including target cells and signaling by receptors NK1.1 and NKp46. To make such a conclusion it would be necessary to dissociate the role of mTOR in NK cell activation and education. For example, had mTOR acute inhibition by Torin not affected NK cell responses to in vitro stimuli, while deletion of mTOR in NK cells in mice still resulted in lack of education, one could then conclude that mTOR may have a role in education.

We thank reviewer 1 for his careful reading of our manuscript and for rightly challenging our interpretation of the data. The dissociation of education and activation process is in our opinion very difficult since the very same molecules are involved in both processes (for example SHP1, SLAM-family molecules, activating and inhibitory receptors…). Considering this limitation, we agree that our data mainly show that mTOR activation is required for NK cell activation through NK activating receptors. We also agree that our data do not implicate mTOR in the education process *per se* but rather that high mTOR activity is the result of the education process. We propose to clarify this point in a revised version of the manuscript. The main messages of the manuscript would then be that 1) *mTOR is a molecular rheostat controlling NK cell activation through NK cell receptors* and that 2) *education signals prime basal mTOR activity.*

Cytokine stimulation, which activates the mTOR pathway, is taken as evidence for a role of mTOR in overcoming the weak responses of uneducated NK cells. This conclusion cannot be made on the basis of the evidence. It is known that IL-2 or IL-15 overcome NK unresponsiveness and stimulate the mTOR pathway. But these cytokines could overcome unresponsiveness independently of their stimulatory effect on the mTOR pathway.

We respectfully disagree with reviewer 1, indeed, in all these experiments described in Figure 7 (previously Figure 6), we show that Torin2, a very specific inhibitor of mTOR abrogates the cytokine effect on Beta2microglobulin KO NK cell reactivity. This clearly shows that mTOR is necessary to overcome the weak response of uneducated NK cells but we agree that it does not show that it is sufficient, a point we could discuss in a revised version of the manuscript. Unfortunately, there is no pharmacologic way to directly and specifically activate mTOR (i.e. without inducing other pathways).

The fact that mTOR contributes to NK activation in the context of IL-2 or IL-15 stimulation does not imply a role in educating NK cells.

We agree that this in vitro experiment on its own “does not imply a role of mTOR in educating NK cells”. As discussed above, we rather propose that high mTOR activity is essential for NK cell reactivity.

This is a nice study on the contribution of the mTOR pathway to NK cell activation both in vitro and for MHC-I negative tumor cell rejection in vivo. Using titrations of inhibitors the authors show that the degree of mTOR activity correlates well with the strength of NK responses, again showing that mTOR is required for NK activation.

We thank reviewer 1 for his careful assessment of our work. We would like to add that this figure also shows that since fine-tuning of mTOR activity results in a corresponding effect on effector functions, the variations detected in the basal activation level of this pathway are physiologically relevant.

mTOR contributes to early steps of NK activation such as calcium responses and activation of integrin.

We would like to stress at this point that this effect of mTOR on proximal signaling was completely unknown and directly connects mTOR activity to two cardinal events of activation that were previously shown to be hallmarks of education in NK cells (Guia et al., 2011; Thomas et al., 2013). These data give a first clue on how fine-tuning of mTOR activity effects downstream activation.

1) The main conclusion is not supported by the data, as described above.

See response to the first comment above.

2) There is no evidence so far that NK cell education by inhibitory receptors is due to a unique educating signal, a hypothesis known as the arming hypothesis. However, there is evidence that continuous stimulation of NK cells leads to hyporesponsiveness. Accordingly, the basis for education is that inhibitory signals prevent induction of hyporesponsiveness. The experiments in this paper should be interpreted in that context. The mTOR pathway could be a particularly sensitive target of activation-induced hyporesponsiveness. In the Introduction section the authors actually say that 'the current viewpoint is that engagement of the NKirs prevents the desensitization of the NK cell due to chronic NKar stimulation'. However, this viewpoint has not been considered in this paper, which is interpreted mainly according to the existence of educating signals.

We agree with the reviewer that we did not show that chronic stimulation through NKar had a negative impact on mTOR basal activity, although we suggested it in the Discussion. We propose to take advantage of a mouse reporter model of activating receptor engagement, the Nur77^GFP^ mice to address this point (Moran et al., 2011). Preliminary analysis of these animals show 1) that GFP levels are increased in NK cells following NKar engagement and 2) that GFP levels are inversely correlated with the expression of educating receptors, ie inversely correlated with the cell education. We will monitor GFP level in Nur77^GFP^ NK cells following transfer in Beta2microglobulin KO mice, a situation that should induce hyporesponsiveness of the transferred cells (Joncker et al., 2010). In parallel, we will follow basal mTOR activity in transferred NK cells.

3) The Discussion is interesting and well written. But it is based on the unsupported premise that mTOR is essential for NK cell education.

We thank reviewer 1 for stating that “the Discussion is interesting and well written”. We have now edited the Discussion according to the response to comment 1.

Reviewer #2:In this manuscript, Yu et al. showed that increased basal activity of the mTOR/Akt pathway is associated with mouse NK cell education defined by expression of the inhibitory receptors. They also found that mTORC1 signaling is regulated by the SHP-1 phosphatase and is associated with the improved responsiveness of educated NK cells. They further showed that upon stimulation through activating NK cell receptors, the mTOR/Akt pathway enhances calcium flux and LFA-1 integrin activation. In accordance with published work, pharmacological inhibition of mTOR results in decreased NK cell reactivity, while acute cytokine stimulation restores responsiveness of uneducated NK cells through mTOR activation.This manuscript studies the previous unaddressed role of mTOR/Akt pathway in NK cell education, and is well written. While most data supports the author's overall conclusion, additional experiments and discussion (as detailed below) are needed to clarify and support all the claims made.

We thank reviewer 2 for his careful reading of our manuscript and we would be happy to provide the requested additional experiments and changes.

Figure 1: What is the phosphorylated protein status between CD11b^lo^, DP, and CD27lo NK cells?

We now present the data in the Figure 1-figure supplement—figure supplement 1.

What is the author's thought on the increased phosphorylation of ERK1/2 T201/Y204 and pNFkB S5356 in NK cells from B2m-deficient mice?

Despite not reaching statistical significance, there is indeed a trend for Beta2microglobulin KO NK cells to display higher level of these phosphorylations. We do not have a definitive answer to that observation. We hypothesize that this increased phosphorylation could be the result of unopposed NKar stimulation in Beta2microglobulin KO mice. As discussed in the response to reviewer 1, preliminary analysis of the Nur77^GFP^ reporter mice indeed show that uneducated NK cells present a higher GFP level suggestive of unopposed chronic stimulation through NKar.

Figure 2: NK cells in Ncr1iCre Mtorfl/fl mice have severe developmental defects. Therefore, can the authors subset NK cells in this figure based on how they subset in Figure 1 (by inhibitor receptors and CD11b and CD27 expression level)?

As a matter of fact, the data presented in Figure 3 (previously Figure 2) are analyzed exactly as in Figure 1, defining the four same subsets using Ly49C and NKG2A combined with CD11b and CD27. The only difference that reviewer 2 might have missed is that we restricted our analysis to the CD27^hi^CD11b^lo^ subset (stated in the figure legend), the only subset present in sufficient number in the NK cell population of Ncr1^iCre^*Mtor*^lox/lox^ mice (Marçais et al., 2014).

Figure 3: The inhibitor used in this figure have different mechanisms of action; some of them inhibit only mTORC1 while some inhibitor both mTORC1 and mTORC2. Can they titrate the amount of the same inhibitor to get the graded pS6 level and compare that with IFN-γ production as well as degranulation?

We have now performed additional experiments to expand the number of different doses of inhibitors and we present the effect of each inhibitor separately in the revised Figure 4 (previously Figure 3) B and C and in Figure 4—figure supplement C. Interestingly, under Rapamycin treatment, where only mTORC1 activity is inhibited, we noticed that the correlation between pAkt and IFNγ/degranulation was not significant anymore suggesting that mTORC2 activity alone is not sufficient to sustain effector functions. We mention it in the revised version of our manuscript.

Figure 6: This figure lacks the control of NK cells from WT B6 mouse treated with IL2 or IL15 with or without Torin2.

We have performed these controls and now showed them in a revised version of Figure 7 (previously Figure 6) as well as in Figure 7—figure supplement 1 and Figure 7—figure supplement 2. As expected, an increased level of mTOR activity in C57BL/6 cells resulted in a commensurate increase in cell reactivity, and this increase could be prevented by co-treatment with Torin2.

Reviewer #3:This group previously published an interesting paper in Nature Immunology reporting that mTOR is required for IL-15 signaling in the development and activation of NK cells. The current manuscript shows that mTOR plays a role in NK cell education; this story is potentially interesting but moreexperiments are necessary to support conclusions.1) In Figure 2, the authors should perform the stimulation of NK cellswith anti-NKP46 to measure% IFNγ+ cells from Ncr1-cre+ and Ncr1-Cre-Mtorlox/lox.

We have now performed the requested experiment. We stimulated the cells using both NKp46 and NK1.1 agonist antibodies and presented the data for reviewer assessment (Author response image 1). As the immature CD27^hi^CD11b^lo^ subset is the only subset present in significant number in the NK cell population of Ncr1^iCre^*Mtor*^lox/lox^ mice (Marçais et al., 2014), we restricted our analysis to it. However, these immature NK cells only secrete low amounts of IFNγ. Moreover, NKp46 levels are decreased in NKp46 Cre/+ mice. Consequently, when immature NK cells from these mice were stimulated through NKp46, no significant IFN-γ production was induced (compare with anti-NK1.1 in Author response image 1). These negative data were thus not included in the revised manuscript.

**Author response image 1. respfig1:** Percentage (mean + SD) of IFNγ^+^ cells among splenic CD11b^lo^ NK cells of the indicated subset from Ncr1^iCre/+^ or Ncr1^iCre/+^ Mtor^lox/lox^ mice following 4-hour stimulation with coated anti-NK1.1 or anti-NKp46 (n=5 mice of each genotype in 3 independent experiments, t-tests comparing each subset in both genotype, **p<0.01, n.s. non significant).

2) In Figure 2, it is unclear how mTOR acts in B2m^-/-^ NK cells. The authors should use as an additional condition: B2m^-/-^+Torin2. Since the authors have mtor conditional ko animals, it would be important to perform a parallel expt using B2m^-/-^ X Ncr1-cre+Mtor lox/lox vs. Ncr1cre+Mtorl ox/lox vs. Ncr1cre+ NK cells

We have performed the Beta2microglobulin KO +/- Torin2 controls and now show them in a revised version of Figure 3 (previously Figure 2) B and C as well as in Figure 3—figure supplement 1. As expected, as the basal activity of mTOR is low but not null in Beta2microglobulin KO NK cells, Torin2 treatment further decrease their reactivity.

3) The authors mentioned that rapamycin primarily inhibits mTORC1 which is true, but only using with a low concentration (e.g. 10 or 20 nM). Concentrations up to 100 nM inhibit both mTORC1 and mTORC2. In Figure 3, the authors used extremely high concentrations of rapalogs, which could lead to some artificial effects. The authors used "3 uM" of KU0063794 for treating NK cells and observed a significant defect of pS6 and pAKT S473. However, other groups have used "100 nM" of KU0063794 to investigate mTOR roles in dendritic cell activation (PMID: 22826320). Thus, authors need to show the cell viability data from those rapalog treatments.

We now present the cell viability in the presence of the different inhibitors in Figure 4—figure supplement 1.

In addition, it is necessary for authors to demonstrate that Ncr1cre+Mtor lox/lox NK cells phenocopy the same results.

We have stimulated NK cells from Ncr1^iCre^ and Ncr1^iCre^*Mtor*^lox/lox^ mice in parallel while measuring the phosphorylation levels of the S6 and Akt proteins and performed the correlation as we did for the C57BL/6 cells treated with inhibitors. We present these results in Figure 4—figure supplement 1.

4) In Figure 5, mTOR regulates glucose metabolism in various cells, but authors only compared glucose uptake by using 2-NBDG from educated and non-educated NK cells; have the authors tried to look at glucose transporter 1 (Glut1) level? Also, it would be nice if the authors could explore in depth other metabolic parameters, for instance mitochondrial function lipid uptake and oxidation?

We have performed these experiments. Concerning Glut1 expression, we ordered an antibody previously used to detect Glut1 expression by flow cytometry (Gerriets et al., 2016). We did not detect any significant difference between reactive and hyporesponsive NK cell population (see Author response image 2). We measured mitochondrial content using Mitotracker green as well as fatty-acid uptake using BodipyFL C16 and both presented differences between reactive and hyporesponsive NK cell populations. These data are now presented in a revised version of Figure 6 (previously Figure 5) B and C. We measured in parallel mitochondrial function using MitoSox Red, lipid droplet content using Bodipy 493/503 and lipid peroxidation using the Click iT Lipid peroxidation kit, but we did not detect any differences (see Author response image 2). This is mentioned in the manuscript as data not shown.

**Author response image 2. respfig2:** Bar graphs showing MFI of Glut1 staining, MitoSox staining, Bodipy incorporation and Lipid peroxidation staining of splenic NK cell subsets from C57BL/6 or β2m^-/-^ mice (mean +SD, n=10 mice of each genotype in 3 independent experiments, except Glut1: 4 mice in 2 independent experiments, no statistical differences were detected between C57BL/6 and β2m^-/-^, t-test). MFI were normalized to the CD11b^lo^ subset of C57BL/6 mice.

5) In Figure 6, it would be great to have data that NK cells from B2m^-/-^+Torin2 in the comparison. If possible it would be nice to see NK cells from B2m^-/-^ X Ncr1cre+ Mtor lox/lox vs. B2m^-/-^ X Ncr1cre+ vs. Ncr1cre+ animals.

We have performed the Beta2microglobulin KO + IL-2 or IL-15 +/- Torin2 controls and now show them in a revised version of Figure 7 (previously Figure 6) A and B as well as in Figure 7—figure supplement 1 and Figure 7—figure supplement 2. For the measure of IFNγ secretion, degranulation and cytotoxicity, the Beta2microglobulin KO +/- Torin2 conditions are presented in Figure 3 (previously Figure 2) B and C alongside C56BL/6 +/- Torin2.

[Editors' note: the author responses to the re-review follow.]

Reviewer #1 had the following specific comments that should be addressed in the re-submission.I appreciate the measured response from the authors and agree it would be fair to let them revise the paper in the way they have proposed. Let me comment on their response.Their replies to specific points are preceded by five points taken out of my summary paragraph. Their reply to the first suggests that I have not delivered my main message clearly enough. The second main message they propose for the revised manuscript is that "education signals prime basal mTOR activity." In the NK field the concepts of priming and education are not well understood and their relationship is unclear. Priming as an early event not known to be reversible. In contrast, education is rapidly reverted by chronic stimulation. There is no evidence that education signals are a prerequisite for acquisition of responsiveness. Therefore, basal mTOR activity may be independent of education signals. However, mTOR activity could be lost or diminished by disarming. The authors should interpret their data within the context of two models: education signals that are required versus an intrinsic ability to respond via mTOR that can be switched off.

We have now profoundly changed the way we present our data to interpret them within the context of the arming vs disarming models. We agree with reviewer 1 that indeed all the experimental evidences collected so far are in favor of the disarming model, i.e. intrinsic reactivity is lost in cells that can not oppose persistent activating stimulation by inhibitory signals. In this context, we now propose that initial activity of the mTOR pathway is set in all NK cells uniformly. This initial activity is then diminished by disarming signals in those NK cells where they are not opposed by inhibitory signals. This is supported by the fact that transfer of reactive NK cells into β2m^-/-^ host results in a loss of mTOR activity concomitant with prolonged NKar stimulation. The main messages of the manuscript are now that 1) *mTOR is a molecular rheostat controlling NK cell activation through NK cell receptors* and that 2) *basal mTOR activity is decreased upon prolonged unopposed activating signal.*

In the second point, I accept their argument on the role of mTOR in cytokine induced education.On the fifth point, I agree that the role of mTOR in early NK activation signals is a new contribution.Specific points: 1. As above. 2. This point relates to my first comment above. The authors have proposed an interesting experiment.

We thank reviewer 1 for judging the proposed experiment as “interesting”. We performed it and found that indeed, transfer of Nur77^GFP^ cells into Beta2microglobulin KO mice is accompanied by 1) a rise in the GFP level indicative of NKar signaling, 2) a loss of cell reactivity upon challenge with agonists of NKar or YAC1 target cells as published by others (Joncker et al., 2010) and 3) a concomitant loss of higher basal phosphorylation of S6 and Akt S473 in NKG2A^+^Ly49C^+^ NK cells. These data are now presented in Figure 2. They support the fact that unopposed NKar stimulation leads to decreased mTOR activity which consequently induces loss of cell reactivity.

Reviewer #2 had the following specific comment that should be addressed in the re-submission.One thing that they still can’t quite explain is why there is increased phosphorylation of ERK1/2 T201/Y204 and pNFkB S5356 in NK cells from B2m-deficient cells. They said that it might be due to unopposed chronic stimulation through NKar… But then if this is the case, NKar also activated mTORC1… What is the difference in the regulation of these phosphorylation in NK cells or why chronic stimulation through NKar negatively impact mTOR signaling?

We still do not have a definitive answer to this last question. We can only hypothesize that the mTOR pathway is particularly sensitive to desensitization. This supposition is based on the previous observations of a number of negative feedback loops desensitizing this pathway (Um et al., 2006). The main but non-exclusive target of these negative feedback loops is a signaling adaptor: the Insulin Receptor Substrate molecule. It is however regulated by the complex interplay of more than 50 S/T phosphorylations (Copps and White, 2012). In addition, whether IRS proteins play a role in lymphocytes is unknown. We thus feel that finding the exact mechanism at play would go beyond the scope of this manuscript, however, we expose this point in the Discussion referring to studies done in the context of the establishment of resistance to insulin.

References

Copps, K.D., and White, M.F. (2012). Regulation of insulin sensitivity by serine/threonine phosphorylation of insulin receptor substrate proteins IRS1 and IRS2. Diabetologia *55*, 2565–2582.

Gerriets, V.A., Kishton, R.J., Johnson, M.O., Cohen, S., Siska, P.J., Nichols, A.G., Warmoes, M.O., de Cubas, A.A., MacIver, N.J., Locasale, J.W., et al. (2016). Foxp3 and Toll-like receptor signaling balance Treg cell anabolic metabolism for suppression. Nat. Immunol. *advance online publication*.

Hukelmann, J.L., Anderson, K.E., Sinclair, L.V., Grzes, K.M., Murillo, A.B., Hawkins, P.T., Stephens, L.R., Lamond, A.I., and Cantrell, D.A. (2015). The cytotoxic T cell proteome and its shaping by the kinase mTOR. Nat. Immunol.

Sarbassov, D.D., Ali, S.M., Sengupta, S., Sheen, J.-H., Hsu, P.P., Bagley, A.F., Markhard, A.L., and Sabatini, D.M. (2006). Prolonged rapamycin treatment inhibits mTORC2 assembly and Akt/PKB. Mol. Cell *22*, 159–68.